# Online Learning and Pricing for Network Revenue Management with Reusable Resources

**Huiwen Jia**
Industrial and Operations Engineering
University of Michigan
Ann Arbor, MI 48109
hwjia@umich.edu

**Cong Shi**
Industrial and Operations Engineering
University of Michigan
Ann Arbor, MI 48109
shicong@umich.edu

**Siqian Shen**
Industrial and Operations Engineering
University of Michigan
Ann Arbor, MI 48109
siqian@umich.edu

## Abstract

We consider a price-based network revenue management problem with multiple products and multiple reusable resources. Each randomly arriving customer requests a product (service) that needs to occupy a sequence of reusable resources (servers). We adopt an incomplete information setting where the firm does not know the price-demand function for each product and the goal is to dynamically set prices of all products to maximize the total expected revenue of serving customers. We propose novel batched bandit learning algorithms for finding near-optimal pricing policies, and show that they admit a near-optimal cumulative regret bound of $\tilde{O}(J\sqrt{XT})$, where $J$, $X$, and $T$ are the numbers of products, candidate prices, and service periods, respectively. As part of our regret analysis, we develop the first finite-time mixing time analysis of an open network queueing system (i.e., the celebrated Jackson Network), which could be of independent interest. Our numerical studies show that the proposed approaches perform consistently well.

## 1 Introduction

Network revenue management (NRM) (Gallego and Van Ryzin 1997) has attained remarkable results in a wide range of applications including airline management, electricity pricing, retailing, healthcare, and leisure (see, e.g., Williamson 1992, Talluri and Van Ryzin 2006, Özer and Phillips 2012). The bulk of existing literature focuses primarily on canonical settings with perishable resources in which the product once sold is permanently removed from inventory (Besbes and Zeevi 2012, Ferreira et al. 2018, Chen and Shi 2019). Little progress has been made for NRM with reusable resources, mainly due to the complexity of analyzing the underlying stochastic systems (typically manifested as stochastic queueing networks). In this paper, we develop easy-to-implement online learning and pricing policies for NRM with reusable resources shared by multiple products and analyze their theoretical and empirical performances.

### 1.1 Problem Description

A service-providing firm is endowed with multiple types of reusable resources, each having a fixed capacity. The firm sells multiple types of products (services), each requiring the use of a sequence

36th Conference on Neural Information Processing Systems (NeurIPS 2022).

of different types of reusable resources (servers). (See, e.g., Figure 2.) Customers randomly arrive according to a price-dependent multivariate Poisson process to request products. Upon requesting one particular product, a customer enters the system by going through a sequence of product-specified reusable resources one by one. During the service process, if there is no available next type of resource upon finishing using the previous type in the sequence, she will join the queue in front of this next type of resource and wait until the next available unit. Both the mean demand rate of each product and the average usage time of each resource depend on the posted price vector. However, such dependence is unknown to the firm *a priori*. We note that under a fixed price vector, the service system can be reduced to the celebrated Jackson Network (Jackson 1963, 1957). The main objective of this paper is to design a "learning-while-doing" algorithm to dynamically post prices of different products, in order to maximize the expected total revenue collected over a finite time horizon.

It is evident that NRM with reusable resources finds many important real-world applications, including cloud computing, hospitality industry, project management, and leisure (Lei and Jasin 2020). For example, firms such as Amazon and Microsoft offer diverse cloud computing services, each requiring a different set of computational resources (virtual machines) for a certain duration of time to finish their computational jobs (see, e.g., Kaewpuang et al. 2013, Püschel et al. 2015).

## 1.2 Main Results and Contributions

We propose new batched bandit algorithms for NRM with reusable resources (RNRM). The performance measure is regret, which is the revenue difference between our learning algorithms and a clairvoyant optimal pricing policy under full information. We use $J, X, T$ to denote the numbers of products, candidate prices, and service periods, respectively. We prove in Theorems 1 and 2 that the cumulative regret is $\tilde{O}(J\sqrt{XT})$, which matches the lower bound along the dimensions of $X$ and $T$ up to a logarithmic factor. It remains open whether one could improve the dependence on $J$. Note that all prior studies on non-reusable settings (e.g., Miao and Wang 2021) show polynomial dependence on $J$ at best. We test diverse instances to show numerical performance of our approaches.

The main contribution of our work is three-fold:

**(Modeling.)** To the best of our knowledge, we are the first to consider the joint learning and optimization problem in NRM with reusable resources. Prior literature on reusable resources only focused on single-product settings (see Jia et al. (2020, 2022) and references therein). It is considerably more challenging to model and analyze the network setting where multiple products share multiple resources (with finite capacities), mainly due to the sequential use of resources (to finish up a service) and the evolving nature of available capacities in real-time. We establish a connection between our model and the celebrated *Jackson Network* (Jackson 1963, 1957).

**(Algorithm.)** We first develop a linear programming (LP) upper bound on the optimal objective value given full information. The LP model provides a static pricing control policy. Therefore, we can cast our RNRM as a multi-armed bandits (MAB) problem where we select among candidate prices (analogs of arms) to minimize the cumulative regret. There are two key challenges. The first lies in the fact that whenever the system posts a new price, the arrival processes change immediately while the system still contains "old" customers from the previously posted price. Therefore, it is difficult to analyze the transient system performance and keep track of the cumulative revenue in a dynamic pricing scheme. We resolve this challenge by proposing batched algorithms. Our algorithms separate the time horizon into successive batches and select a price for each batch. We further divide each batch into two intervals. In the first interval, the system completes serving the previous customers and reaches the steady-state under the new price. In the second interval, the system maintains the steady-state (exploiting the new price). This batched framework not only allows for system stabilization (under new prices) but also allows for infrequent $O(\log T)$ price changes, which is more implementable in practical settings (see Cheung et al. 2017). The second challenge is more mathematical because we must ensure that the regret does not scale exponentially in the number of products or resources (given the combinatorial nature of the underlying graph).

**(Regret Analysis.)** To cope with the first challenge mentioned above, we bound the *loss of nonstationarity* by bounding the length of the first interval of each batch. We prove that the system can reach the new steady-state reasonably fast (see mixing times of Jackson Network in Propositions 3 and 4). To the best of our knowledge, this is the first result giving a finite-time bound on mixing times of Jackson Networks. To cope with the second challenge, we decompose the resulting Jackson Network into

topological layers and present a novel nested coupling argument (based on the layers). The coupling argument of Jia et al. (2020) in the single-resource single-product setting cannot be directly applied to our network setting. First, in the network of M/M/c queues, the arrival and waiting processes depend on previously visited resources, which is significantly more complex to analyze upon price changes. To resolve this challenge, we have to carefully construct virtual and alternative queues to explicitly account for such dependency. On a high level, the difficulty lies in that the original queues have a time-varying rate from an initial rate to a stable rate upon any price change. Given a carefully chosen time period $\tau$, we first construct alternative queues that start from zero initial rate and ramp up directly to the stable rate at a timestamp before $\tau$ so that the customer distribution matches the original queues in period $\tau$. We then construct virtual queues that could clear all previously arrived customers at a specific time $\tau$, and show that their mixing times provide an upper bound on the alternative queues and consequently the original queues. Second, even ignoring the subtleties of time-varying rates (upon price changes), if one applies the coupling argument of Jia et al. (2020) in a brute-force way to every resource in the network, the coupling probability would result in an undesirable exponential scaling in the number of products. We decompose the network into topological layers so that the resulting coupling probability only scales linearly in the number of products.

## 1.3 Literature Review

**NRM with Reusable Resources and Demand Learning.** Network, demand learning, and reusable resources are three key elements of this study. We only review related literature that shares at least two of the three elements.

*NRM with demand learning.* Due to the easy accessibility of real-time sales data, there have been recent studies on developing learning and pricing algorithms with probably near-optimal regret bound for NRM (see, e.g., Besbes and Zeevi 2012, Chen et al. 2019, Chen and Shi 2019, Badanidiyuru et al. 2018, Agrawal and Devanur 2016, Ferreira et al. 2018). All the aforementioned study considers the canonical settings (first defined by Gallego and Van Ryzin 1997), i.e., non-reusable resources.

*NRM with reusable resources.* The only existing studies on RNRM are Owen and Simchi-Levi (2018), Rusmevichientong et al. (2020) and Lei and Jasin (2020). Lei and Jasin (2020) considered a deterministic service time assumption, and developed an asymptotically near-optimal real-time pricing control algorithm. Owen and Simchi-Levi (2018) considered a joint assortment and pricing problem under a multi-resource setting and provided constant factor performance guarantees. Rusmevichientong et al. (2020) considered a dynamic assortment and pricing problem with reusable products and proposed a provably near-optimal policy based on approximate dynamic programming.

*Demand learning with reusable resources.* The aforementioned literature on reusable resources all assumed known distributional information of the underlying model. There has been little work considering demand learning for service systems with reusable resources due to complex underlying system dynamics. The only studies that considered demand learning for reusable resources are Chen et al. (2020), Jia et al. (2020, 2022), and all considered only the single resource setting. Chen et al. (2020) proposed an online learning framework for a dynamic pricing and capacity sizing problem in a GI/GI/1 queue and Jia et al. (2020, 2022) developed MAB-based algorithms for optimal dynamic pricing in an M/M/c queue for the single product single resource setting.

To the best of our knowledge, we are the first to thoroughly study all three elements.

**Transient Analysis of Jackson Networks.** Jackson Network has received much attention among the queueing networks because its steady-state distribution is particularly simple to compute as a product-form solution (Jackson 1963, 1957). The queue lengths across the nodes in Jackson Networks, i.e., a network of M/M/c queues, during the transient state can be bounded via techniques of stochastic dominance (see Massey 1986, 1984, 1987). However, how the network reacts to system changes remains an open question (see, e.g., Mohamed et al. 2003, 2005). Kelton and Law (1985) pointed out that a queueing system needs to run "long-enough" for removing the effects brought by the initial starting point and then reach an ensured "steady-state" portion of the run. The only work that explicitly developed a finite-time bound on mixing times of a queueing system is a recent paper by Jia et al. (2020). In contrast, our paper gives the first finite-time high probability bound on mixing times of Jackson Networks (a network consisting of M/M/c queues) upon action changes.

**Batched Bandit Approaches.** UCB2 (Auer et al. 2002) and Improved-UCB (Auer and Ortner 2010) were implicitly $M$-batch policies with $M = \Theta(\log(T))$. Later, Perchet et al. (2016) studied batched

MAB in the two-armed case under a static batch design and Gao et al. (2019) extended the algorithms to the general multiple-armed case and adaptively determined batch size. There is also rich literature focusing on batched MAB problems with switching cost. For example, Cesa-Bianchi et al. (2013) considered uniform switching cost in the objective and Simchi-Levi and Xu (2019) treated the finite switching cost budget as a hard constraint. In contrast, we construct the batched action framework to control the loss of the nonstationarity (resulted from the transient time after each action change).

## 2 Problem Formulation

Consider a firm that manages $\mathcal{I}$ types of reusable resources to provide $\mathcal{J}$ types of products over a finite time horizon $T$. Denote the capacity of resource type $i \in \mathcal{I}$ as $c_i$. At the beginning of each period, the firm posts a price $\mathbf{x} = \{x_j : j \in \mathcal{J}\}$ for all products from a given price candidate set $\mathcal{X} \subset \mathbb{R}^{|\mathcal{J}|}$. Let $I = |\mathcal{I}|$, $J = |\mathcal{J}|$, and $X = |\mathcal{X}|$. Each arriving customer requests one product $j \in \mathcal{J}$, which is processed through an ordered sequence of resources. Customers arrive following a price-dependent multivariate Poisson process with a total rate $\alpha(\mathbf{x})$, which is unknown to the decision maker *a priori*. The service time of resource $i$ follows an exponential distribution regardless of which product type it belongs to, which also depends on posted price $\mathbf{x}$. If there is not enough capacity for the current resource, the customer will join the queue of this resource until being served. The resources are released to process the next customers upon finishing serving the current ones.

We consider two popular payment models, the lump sum payment model (presented in the main manuscript) and the service duration-dependent payment model (presented in Appendix F).

- (**Lump sum payment model.**) A customer of product $j \in \mathcal{J}$ pays a lump sum $x_j$ to the firm. More elaborately, if a customer chooses product $j$, then the customer will pay $x_j$ to the firm regardless of how long the services over resources are. In our illustrative example, each product can be quoted a fixed price (i.e., not service duration dependent), which could be seen as a preset "cloud solution package". The customers may prefer or benefit from this fixed price (contract) from their own accounting and cash flow management. Under this model, even though the customers are not sensitive to service durations, the cloud platform is nevertheless sensitive to service durations since they affect how many products/services can be completed within a finite horizon.

- (**Service duration dependent payment model.**) A customer of product $j \in \mathcal{J}$ pays the amount that is equal to the per-unit-time $x_j$ multiplied by the total service duration. This payment scheme is duration dependent, which is also ubiquitous, especially in cloud computing applications (e.g., how much time is spent using EC2 in our illustrative example). In the interest of space, the model and analytical details are presented in Appendix F.

Consider a resource-consumption matrix $A_{ji} \in \mathbb{R}^{J \times I}$, where element $a_{ji} = 1$ indicates that service type $j$ requires to be processed on server type $i$. We use an $I$-dimensional vector, $\mathbf{y} \in \{0, 1, \ldots, \infty\}^I$ to denote the number of customers across different types of resources. The firm aims at finding a pricing policy as $\pi : \{(\mathbf{y}, t) : \mathbf{y} \in \{0, 1, \ldots, \infty\}^I, t = 1, \ldots, T\} \to \mathcal{X}$, where the firm selects price $\pi(\mathbf{y}, t) \in \mathcal{X}$ for period $t$ when there are $\mathbf{y}$ customers in the system at the beginning of period $t$, to maximize the expected cumulative revenue $Y^\pi$.

**Assumption 1.** *("Turn-off" price.) For each product $j \in \mathcal{J}$, there exists a "turn-off" price $\bar{p}_j < \infty$ such that the demand for this product is zero when the posted price is larger than this turn-off price.*

Assumption 1 is widely assumed in NRM studies (see, e.g., Lei and Jasin 2020). This turn-off price can be used to effectively turn off the demand for undesired products whenever needed. Therefore, for each candidate price $\mathbf{x} \in \mathcal{X}$, we can further decompose the products into turn-on product set $\mathcal{J}_{on}(\mathbf{x})$ and a turn-off product set $\mathcal{J}_{off}(\mathbf{x})$, where the latter can be empty.

**Assumption 2.** *There exists an order of server types $\mathcal{I}$ such that the processing sequence of servers $i_1, i_2, \ldots, i_n$ always has $i_1 < i_2 < \ldots < i_n$ for any product $j \in \mathcal{J}$.*

One can understand Assumption 2 via viewing a network. Consider a graph with resources as nodes and draw a directed arc from node $i$ to $i'$ if any product requires successive processing on resources $i$ and $i'$. If the system satisfies Assumption 2, then this graph is acyclic, i.e., no cycles, and there exists at least one topological order such that all arcs are from a smaller index to a larger index. Without

loss of generality, we consider one such topological order when referring to resource $i$ in the rest part of this paper. To better utilize this network representation, we also add a source node $s$ to represent customers entering the system and a sink node $t$ to represent customers exiting the system. Thus, the consumption matrix can be extended to $A_{ji} \in \mathbb{R}^{J \times (I+2)}$ and $a_{js} = a_{jt} = 1$ for all products.

**Assumption 3.** *(Product-form demand.)* *Any given price vector $\mathbf{x}$ induces a (Markovian) routing matrix $P(\mathbf{x})$ where its $(i, i')$th entry $p_{i,i'}$ represents the product demand portion who will consume resource $i'$ next out of those who consume resource $i$. Then the demand rate for any product $j \in \mathcal{J}$, represented by a sequence of resources $s, i_1, i_2, \ldots, i_n, t$, is given by*

$$\alpha_j = \alpha(\mathbf{x}) p_{s,i_1} p_{i_1,i_2} \ldots p_{i_{n-1},i_n} p_{i_n,t},$$

*which means that the demand rate for product $j$ is equal to the total demand rate $\alpha(\mathbf{x})$ times a sequence of routing probabilities of $P(\mathbf{x})$, under posted price vector $\mathbf{x}$.*

We use Figure 2 in the Appendix to demonstrate a concrete example. With Assumptions 2 and 3, the service system under any posted price $\mathbf{x} \in \mathcal{X}$ can be reduced to a Jackson Network consisting with $I$ number of M/M/c queues (when the system only contains customers arriving under this price, i.e., after finishing serving other customers arriving under the previously posted prices). The main idea is that there exists a mapping $\mathbf{x} \mapsto \alpha(\mathbf{x}), P(\mathbf{x})$ that gives rise to product demand pattern. Note that our assumptions are mild in the sense that this mapping can be nonparametric. The decision maker does not know this underlying mapping and needs to learn it while maximizing revenue on the fly.

Given any price $\mathbf{x}$, (i) on one hand, this price affects the total demand $\alpha(\mathbf{x})$ entering the system. (ii) On the other hand, this price affects the internal competition between products, which is reflected in the server routing matrix $P(\mathbf{x})$. As a result, $\alpha(\mathbf{x})$ and $P(\mathbf{x})$ define a Jackson Network well and the demand rate $\alpha_j(\mathbf{x})$ of each product type is also determined (use vector $\vec{\alpha}_{\mathbf{x}} \in \mathbb{R}^J$ to denote demand of products). We use $\lambda_i(\mathbf{x})$ and $\mu_i(\mathbf{x})$ to denote the demand and service rates of resource type $i$.

**Assumption 4.** *(Stability.)* *The queueing system of each resource is stable under any price, i.e., $c_i \mu_i(\mathbf{x}) > \lambda_i(\mathbf{x})$. Therefore, the Jackson Network is stable as well.*

Assumption 4 ensures that the underlying Jackson Network is stable under any candidate price (i.e., the queue length will not grow indefinitely). This assumption can be readily satisfied by putting appropriate lower bounds on the price vector.

**Assumption 5.** *(Change of rates.)* *For any node, consider the arrival and service rates $(\lambda_1, \mu_1)$ and $(\lambda_2, \mu_2)$ under $\mathbf{x}_1, \mathbf{x}_2 \in \mathcal{X}$ where $\lambda_1 \geq \lambda_2$, also denote the utilization factor by $\rho_1 = \lambda_1/\mu_1$ and $\rho_2 = \lambda_2/\mu_2$, respectively, then 1) $\lambda_1 - \lambda_2 \leq \frac{\rho_1}{-3e \log \rho_1}$ and 2) if $\mu_1 > \mu_2$, then $\rho_1 \geq \rho_2^2$.*

This assumption is used to justify Lemmas 3 and 4 (used in the proof of Propositions 3 and 4) and the original proofs are in Jia et al. (2020). These conditions are mild and easy to validate in practice in the following sense. 1) The difference in arrival rates between two prices is bounded (by a large constant). This can be easily satisfied by interpolating intermediate prices in the candidate price set $\mathcal{X}$. 2) If both the service and arrival rate under a price is higher than or equal to another price, the utilization factor in the former case must be greater than the latter squared. A simple sufficient condition to guarantee $\rho_1 \geq \rho_2^2$ is $\rho_1 \geq \rho_2$, which reduces to $\lambda_1/\lambda_2 \geq \mu_1/\mu_2$, and thus the practitioners only need to validate that the ratio of change in arrival rates exceeds the ratio of change in service rates.

**Assumption 6.** *We assume $\log T \geq 4$ (to ensure sufficient time for mixing).*

## 3 Regret, Relaxed Regret, and Static Price Benchmark

Under the full information scenario, where the firm knows the underlying arrival rate $\alpha_j(\mathbf{x})$ of all products $j \in \mathcal{J}$ under any candidate price $\mathbf{x} \in \mathcal{X}$. We denote a state-dependent optimal policy as $\pi^*$, where $\pi^* = \arg\max_\pi Y^\pi$, and let $Y^* = Y^{\pi^*}$. Therefore, the regret of any pricing policy $\pi$ can be defined as $\text{Regret}(\pi, T) = Y^* - Y^\pi$. In short, we aim at finding heuristic pricing policies that lead to a small regret. However, finding the state-dependent optimal pricing policy $\pi^*$ is computationally intractable even under the full information scenario. Firstly, it requires solving a dynamic program with an infinite number of potential states, in the form of $(\mathbf{y}, t)$, which suffers from the curse of dimensionality. Moreover, the transient dynamics of a queueing network are complex and hard to analyze especially when the offered price changes over periods. Thus, it is impossible to obtain $\pi^*$

and $Y^*$. To tackle this problem, we develop an upper bound on the expected revenue achievable by the state-dependent optimal expected revenue (see a similar approach in Jia et al. 2020).

**An LP based upper bound.** Consider continuous decision variables $\pi_{\mathbf{y}t}^{\mathbf{x}}$, $a_{\mathbf{y}t}$, and $Y_{\mathbf{y}t}$ for $\mathbf{y} \in \{0, 1, \ldots, \infty\}^I$, $t = 1, \ldots, T$, and $\mathbf{x} \in \mathcal{X}$. We formulate a linear program (LP) as follows. We also discuss the sufficiency of using $(\mathbf{y}, t)$ as the state variable in Appendix C.

$$Y^{\text{LP}} = \max_{\pi, a, Y} \sum_{\mathbf{y} \in \{0,1,\ldots,\infty\}^I} \sum_{t=1}^{T} a_{\mathbf{y}t} Y_{\mathbf{y}t} \tag{1a}$$

$$\text{s.t.} \sum_{\mathbf{y} \in \{0,1,\ldots,\infty\}^I} a_{\mathbf{y}t} = 1, \quad \forall t = 1, \ldots, T \tag{1b}$$

$$\sum_{\mathbf{x} \in \mathcal{X}} \pi_{\mathbf{y}t}^{\mathbf{x}} = 1 \quad \forall \mathbf{y} \in \{0, 1, \ldots, \infty\}^I, \ \forall t = 1, \ldots, T \tag{1c}$$

$$Y_{\mathbf{y}t} \leq \sum_{\mathbf{x} \in \mathcal{X}} \pi_{\mathbf{y}t}^{\mathbf{x}} \cdot \vec{\alpha}_{\mathbf{x}}^T \mathbf{x} \quad \forall \mathbf{y} \in \{0, 1, \ldots, \infty\}^I, \ \forall t = 1, \ldots, T \tag{1d}$$

$$0 \leq a_{\mathbf{y}t} \leq 1 \quad \forall \mathbf{y} \in \{0, 1, \ldots, \infty\}^I, \ \forall t = 1, \ldots, T \tag{1e}$$

$$0 \leq \pi_{\mathbf{y}t}^{\mathbf{x}} \leq 1 \quad \forall \mathbf{y} \in \{0, 1, \ldots, \infty\}^I, \ \forall t = 1, \ldots, T, \ \forall \mathbf{x} \in \mathcal{X}. \tag{1f}$$

**Proposition 1.** *The LP solution provides an upper bound on the optimal revenue, i.e., $Y^* \leq Y^{LP}$, and $Y^{LP} = T \cdot \vec{\alpha}_{\tilde{\mathbf{x}}}^T \tilde{\mathbf{x}}$ where $\tilde{\mathbf{x}} = \arg\max_{\mathbf{x} \in \mathcal{X}} \vec{\alpha}_{\mathbf{x}}^T \mathbf{x}$.*

*Proof Sketch.* Consider an arbitrary policy $\hat{\pi}$. Let variable $\hat{a}_{\mathbf{y}t}$ take the value of probability that the system has $\mathbf{y}$ customers at the beginning of period $t$ for $\mathbf{y} \in \{0, 1, \ldots, \infty\}^I$ and $t = 1, \ldots, T$. Let variable $\hat{\pi}_{\mathbf{y}t}^{\mathbf{x}} = 1$ if the policy $\hat{\pi}$ chooses price $\mathbf{x}$ at state $(\mathbf{y}, t)$, and 0 otherwise. Let variable $\hat{Y}_{\mathbf{y}t}$ take the value of expected revenue collected before the end of period $T$ from the customers who arrive during period $t$, when the system has $\mathbf{y}$ customers at the beginning of period $t$. Then we can show that the decision variables $(\pi, a, Y)$ associated with any admissible policy must satisfy the constraints of the linear program (1), and the expected revenue of this admissible policy is exactly the corresponding objective value in (1). We can readily check that $T \cdot \vec{\alpha}_{\tilde{\mathbf{x}}}^T \tilde{\mathbf{x}}$ is an upper bound of $Y^{\text{LP}}$. We can construct a solution which can obtain this upper bound and thus $Y^{\text{LP}} = T \cdot \vec{\alpha}_{\tilde{\mathbf{x}}}^T \tilde{\mathbf{x}}$. $\square$

**Definition 1.** *The relaxed regret is defined as $\overline{Regret}(\pi, T) = Y^{LP} - Y^{\pi}$.*

Based on Proposition 1, we have for any policy $\pi$, the corresponding relaxed regret is a valid upper bound of its regret, i.e., $\overline{\text{Regret}}(\pi, T) \geq \text{Regret}(\pi, T)$, and it suffices to bound the relaxed regret.

## 4 Batched Bandit Algorithms

The basis for implementing an Upper Confidence Bound (UCB) type of algorithm is identifying an upper confidence bound of the expected collected revenue of each candidate price. We start with the steady-state revenue rate. We can show that this steady-state revenue rate can be used to compute a lower bound of the expected revenue during consecutive periods (see proof of Theorem 1). Therefore, it is equivalent to deriving an upper confidence bound of the steady-state revenue rate based on historical sales information. Jia et al. (2020) established a concentration bound of the steady-state revenue rate under the single product setting. Here we develop a concentration bound for the multi-product multi-resource setting.

We use $n_j(\mathbf{x})$ to denote the number of arrived customers requesting product $j$ under price $\mathbf{x}$. We use $\hat{d}_{jk}(\mathbf{x})$ to denote the observed time interval between the arrival time of customer $k$ and the arrival time of customer $k - 1$ who are requesting product $j$ under the price $\mathbf{x}$. We use $\bar{d}_j(\mathbf{x})$ to denote the empirical mean of the inter-arrival time and thus $\bar{d}_j(\mathbf{x}) = \sum_{k=1}^{n_j(\mathbf{x})} \hat{d}_{jk}(\mathbf{x}) / n_j(\mathbf{x})$, for $j \in \mathcal{J}$.

**Proposition 2.** *For any price $\mathbf{x} \in \mathcal{X}$, if the number of arrival time observations for each turn-on product $n_j(\mathbf{x}) \geq 8 \log(J^{\frac{1}{4}} T)$ for $j \in \mathcal{J}_{on}(\mathbf{x})$, then we have*

$$\mathbb{P}\left(\left|\vec{\alpha}_{\mathbf{x}}^T \mathbf{x} - \sum_{j \in \mathcal{J}_{on}(\mathbf{x})} \frac{x_j}{\bar{d}_j(\mathbf{x})}\right| \leq \sum_{j \in \mathcal{J}_{on}(\mathbf{x})} \frac{x_j \sqrt{32 \log(J^{\frac{1}{4}} T)}}{\bar{d}_j(\mathbf{x}) \sqrt{n_j(\mathbf{x})}}\right) \geq 1 - \frac{2}{T^4}.$$

The proof of Proposition 2 is based on the concentration inequality for sub-exponential random variables (see, e.g., Boucheron et al. 2013, Jia et al. 2020) and is provided in Appendix C. Note that $\vec{\alpha}_{\mathbf{x}}^T \mathbf{x}$ is the steady-state revenue rate and denote it by $r(\vec{\alpha}_{\mathbf{x}}, \mathbf{x})$. Proposition 2 says that with a certain amount of arrival time observations of turn-on products under a candidate price, we can derive a valid upper bound of the steady-state revenue rate with a high probability guarantee. Hence, we define the corresponding upper and lower confidence bounds of the stationary revenue rate associated with price $\mathbf{x}$ by the end of period $t$ as follows.

**Definition 2.** *Define* $\mathbf{Rad}_t(\mathbf{x}) = \sum_{j \in \mathcal{J}_{on}(\mathbf{x})} \frac{x_j \sqrt{32 \log(J^{\frac{1}{4}} T)}}{\bar{d}_j(\mathbf{x}) \sqrt{n_j(\mathbf{x})}}$ *as the confidence radius by the end of period* $t$. *The upper and lower confidence bounds of the revenue rate associated with price* $\mathbf{x}$ *are:*

$$U_t(\mathbf{x}) = \sum_{j \in \mathcal{J}_{on}(\mathbf{x})} \frac{x_j}{\bar{d}_j(\mathbf{x})} + \mathbf{Rad}_t(\mathbf{x}), \qquad L_t(\mathbf{x}) = \sum_{j \in \mathcal{J}_{on}(\mathbf{x})} \frac{x_j}{\bar{d}_j(\mathbf{x})} - \mathbf{Rad}_t(\mathbf{x}). \tag{2}$$

### 4.1 RNRM-UCB

With Definition 2, we can leverage the idea of UCB to solve RNRM. We separate $T$ periods into two phases. The first phase is the *Warm-up* Phase, where we offer each candidate price one by one until we collect $8 \log(J^{\frac{1}{4}} T)$ number of arrival time intervals for each turn-on product $j \in \mathcal{J}_{on}(\mathbf{x})$ under each price. Therefore, the total length of this Warm-up Phase is $O(X \log(J^{\frac{1}{4}} T))$. After this Warm-up Phase, Proposition 2 is valid to use. The second phase is the *Learning* Phase. The UCB algorithm cannot be directly used because in order to estimate the expected revenue when offering a price, we have to wait until the system reaches the steady-state. The key idea is to reduce the number of price changes and further reduce the intractable transient state performance. We separate the Learning Phase into $M$ number of batches. Define $I_m = 2^m$, $\forall m = 1, \ldots, M$. Each batch $m$ contains $I_m L \tau$ periods, where $\tau = (\log T)^2$ and $L$ is the number of layers defined in Definition 3. Therefore, the batch size is exponentially increasing. We can then add $m$ in the subscript of notation $(n_{mj}(\mathbf{x}), \hat{d}_{jk}(\mathbf{x}), \bar{d}_{mj}(\mathbf{x}), \mathbf{Rad}_m(\mathbf{x}), U_m(\mathbf{x}), L_m(\mathbf{x}))$ to represent their values by the end of batch $m$. At the beginning of each batch $m + 1$, RNRM-UCB chooses price $\mathbf{x}$ with the highest $U_m(\mathbf{x})$ and offers this price during batch $m + 1$. We present the algorithmic details in Algorithm 1.

---
**Algorithm 1** RNRM-UCB.
---
1: Input: $T$, $\mathcal{X}$, service network, turn-off prices.
2: Initialize: $L$, $\tau$, $I_m$, $M$, $m = 0$.
3: Warm-up Phase:
4: **for** $\mathbf{x} \in \mathcal{X}$ **do**
5:     Offer $\mathbf{x}$, record inter-arrival time $\hat{d}_{jk}(\mathbf{x})$ for arriving customers of product $j \in \mathcal{J}_{on}(\mathbf{x})$.
6:     **if** $n_{mj}(\mathbf{x}) \geq 8 \log(J^{1/4} T)$ for $j \in \mathcal{J}_{on}(\mathbf{x})$ **then**
7:         Continue.
8:     **end if**
9: **end for**
10: Compute and update UCB for all prices as defined in (2).
11: Learning Phase:
12: **for** $m = 1, \ldots, M$ **do**
13:     Choose $\mathbf{x}_m = \text{argmax}_{\mathbf{x} \in \mathcal{X}} U_{m-1}(\mathbf{x})$.
14:     Offer $\mathbf{x}_m$ in batch $m$, i.e., for $I_m L \tau$ periods.
15:     Record inter-arrival time $\hat{d}_{jk}(\mathbf{x}_m)$ for arriving customers of product $j \in \mathcal{J}_{on}(\mathbf{x}_m)$.
16:     Compute and update UCB of $\mathbf{x}_m$ as defined in (2).
17: **end for**
---

### 4.2 RNRM-TS

Considering the firm holds prior information about the rates, then we can add the Bayesian assumption to this RNRM problem. By the conjugate prior theory (see, e.g., Minton et al. 1962, Diaconis and Ylvisaker 1979), we assume that RNRM-TS holds Gamma priors on $\alpha_j(\mathbf{x})$ of product $j \in \mathcal{J}_{on}(\mathbf{x})$, $\forall \mathbf{x} \in \mathcal{X}$ and the posteriors are updated by the end of batch $m$, $\forall m = 1, \ldots, M$. Specifically, we

set the prior distribution of $\alpha_j(\mathbf{x})$ as **Gamma**$(\eta^{\alpha_j(\mathbf{x})}, \beta^{\alpha_j(\mathbf{x})})$. After observing one arrival time interval $\hat{d}_{jk}(\mathbf{x})$ of product $j$ while offering price $\mathbf{x}$, the posterior distribution of $\alpha_j(\mathbf{x})$ is computed as **Gamma**$(\eta^{\alpha_j(\mathbf{x})} + 1, \beta^{\alpha_j(\mathbf{x})} + \hat{d}_{jk}(\mathbf{x}))$. To implement RNRM-TS, at the beginning of batch $m$, we sample parameters $\hat{\alpha}_j(\mathbf{x})$ for product $j \in \mathcal{J}_{\text{on}}(\mathbf{x})$, $\forall \mathbf{x} \in \mathcal{X}$, following the updated posterior distributions. Then we offer the price with the maximum sampled stationary network revenue rate $r(\hat{\vec{\alpha}}_{\mathbf{x}}, \mathbf{x}) = \sum_{j \in \mathcal{J}_{\text{on}}(\mathbf{x})} \mathbf{x}_j \hat{\alpha}_j(\mathbf{x})$. We present the algorithmic details of RNRM-TS in Algorithm 2.

---

**Algorithm 2** RNRM-TS.

---

1: Input: $T$, $\mathcal{X}$, service network, turn-off prices, $\eta^{\alpha_j(\mathbf{x})}$, $\beta^{\alpha_j(\mathbf{x})}$, $\forall j \in \mathcal{J}_{\text{on}}(\mathbf{x})$, $\forall \mathbf{x} \in \mathcal{X}$.
2: Initialize: $L$, $\tau$, $I_m$, $M$, $m = 0$.
3: Warm-up Phase:
4: **for** $\mathbf{x} \in \mathcal{X}$ **do**
5:      Offer $\mathbf{x}$, record inter-arrival time $\hat{d}_{jk}(\mathbf{x})$ for arriving customers of product $j \in \mathcal{J}_{\text{on}}(\mathbf{x})$.
6:      **if** $n_{mj}(\mathbf{x}) \geq 8\log(J^{1/4}T)$ for $j \in \mathcal{J}_{\text{on}}(\mathbf{x})$ **then**
7:          Continue.
8:      **end if**
9: **end for**
10: Update $\eta^{\alpha_j(\mathbf{x})}$, $\beta^{\alpha_j(\mathbf{x})}$, $\forall j \in \mathcal{J}_{\text{on}}(\mathbf{x})$, $\forall \mathbf{x} \in \mathcal{X}$.
11: Learning Phase:
12: **for** $m = 1, \ldots, M$ **do**
13:      Sample $\hat{\alpha}_j(\mathbf{x}) \sim$ **Gamma**$(\eta^{\alpha_j(\mathbf{x})}, \beta^{\alpha_j(\mathbf{x})})$ for product $j \in \mathcal{J}_{\text{on}}(\mathbf{x})$, $\forall \mathbf{x} \in \mathcal{X}$
14:      Choose $\mathbf{x}_m = \text{argmax}_{\mathbf{x} \in \mathcal{X}} r(\hat{\vec{\alpha}}_{\mathbf{x}}, \mathbf{x})$.
15:      Offer $\mathbf{x}_m$ in batch $m$, i.e., for $I_m L \tau$ periods.
16:      Record inter-arrival time $\hat{d}_{jk}(\mathbf{x}_m)$ for arriving customers of product $j \in \mathcal{J}_{\text{on}}(\mathbf{x}_m)$.
17:      Compute and update $\eta^{\alpha_j(\mathbf{x}_m)}$, $\beta^{\alpha_j(\mathbf{x}_m)}$, $\forall j \in \mathcal{J}_{\text{on}}(\mathbf{x}_m)$.
18: **end for**

---

# 5 Performance Analysis

## 5.1 High Probability Bound for Mixing Time

When the firm starts service from an empty state, i.e., zero customers in the network service system, or switches to another price from the currently offered price, the service (queueing network) system enters a transient state and needs a certain amount of time to reach the steady-state again (referred to as the mixing time). Recall that the service system can be reduced to a Jackson Network under a specifically offered price vector, upon completion of existing customers under the previously posted price. Therefore, we evaluate how quickly a Jackson Network can reach the steady-state. To the best of our knowledge, this is the first result giving a finite-time high probability bound on mixing times of Jackson Network system (upon action changes). Technically, we develop a new coupling argument between this target system and a virtual system starting from a state sampled from the steady-state distribution and maintaining the steady-state thereafter.

**Poisson arrival and departure under steady-state.** Under the steady-state of an M/M/c queue (if it exists, i.e., $\rho = \lambda/c\mu < 1$), then the process following which the customers leave the system is also a Poisson process with rate $\lambda$. Therefore, one can easily derive that the arrival as well as the departure processes of each type of resource (each node in the network representation) are Poisson processes with the same rate $\lambda_i$ under the steady-state (see the celebrated Burke's Theorem in Burke 1956).

Jia et al. (2020) is the only work for studying finite-time bounds on the mixing time for an M/M/c queue. However, their results cannot be directly applied here because (i) the arrival process of each resource is not a well-defined Poisson process and (ii) the starting point of the later queues is not zero state nor a steady-state of old system parameters. Even with these two challenges tackled, this approach will induce a mixing time which is linear on the number of resources. This result is not desired for systems with multiple (especially large) resources. We develop a new mixing time analysis for Jackson Network. We divide the resources of the service network into layers and derive a high-probability mixing time that is only linear on the number of layers.

**Definition 3.** *We divide the nodes (types of servers) into layers indexed by the maximum number of previously occupied nodes (including the virtual source node) along the service routes of all products. For example, Nodes 1 and 2 of the illustrative example (see Figure 2) is Layer-1 nodes, and Node 3 is a Layer-2 node. We denote the number of layers by $L$ and the set of Layer-$l$ nodes as $\mathcal{I}_l$.*

**Proposition 3.** *(Coupling probability of a Jackson Network starting from empty state)* *Define $\mathcal{A}_t$ to be the event in which the Jackson Network reaches the steady-state by time $t$. For $t \geq L\tau$, where $\tau = (\log T)^2$, we have $\mathbb{P}(\mathcal{A}_t) \geq 1 - \frac{2I}{T^2}$.*

*Proof Sketch.* We use the aforementioned Poisson arrival and departure property of the nodes in a Jackson Network to bound the mixing time of the nodes layer by layer. We define events $\mathcal{A}_{t',l} = \{$all layer-l nodes reach the steady-state by time $t'\}$ and $\mathcal{A}_{t'}^i = \{$the node $i$ reaches the steady-state by time $t'\}$. By treating the nodes in the same layer together,

$$\mathbb{P}(\mathcal{A}_t) = \mathbb{P}(\mathcal{A}_{L\tau,1}\mathcal{A}_{L\tau,2}\mathcal{A}_{L\tau,3}\dots\mathcal{A}_{L\tau,L}) \geq \mathbb{P}(\mathcal{A}_{\tau,1}) \prod_{l=2}^{L} \mathbb{P}(\mathcal{A}_{l\tau,l}|\mathcal{A}_{\tau,1}\dots\mathcal{A}_{(l-1)\tau,l-1}).$$

We then prove that $\mathbb{P}(\mathcal{A}_{l\tau,l}|\mathcal{A}_{\tau,1}\dots\mathcal{A}_{(l-1)\tau,l-1}) \geq 1 - (2I_l)/T^2$, where $I_l = |\mathcal{I}_l|$, by establishing a coupling argument between the target queue (with the help of an artificially constructed alternative queue) and a virtual queue starting from a state sampled from the steady-state distribution and maintaining the steady-state thereafter. The detailed proof is provided in Appendix E. $\square$

In addition, the transient state of the Jackson Network, as well as each resource queue (or each node), can be induced not only by starting from empty state but also by changing the price from the previously posted price. It is possible that when a new price is posted, the customers who arrived under the previously posted price still remain in the system. Therefore we also need to ensure that the coupling also occurs with high probability when price changes.

**Proposition 4.** *(Coupling probability of Jackson Network starting from steady-state of the previous price.)* *Define $\mathcal{B}_t$ to be the event in which the Jackson Network reaches the steady-state by time $t$. For $t \geq 2L\tau$, where $\tau = (\log T)^2$, we have $\mathbb{P}(\mathcal{B}_t) \geq 1 - \frac{4I}{T^2}$.*

*Proof Sketch.* This proposition is derived based on Proposition 3 and Assumption 5. We define $\mathcal{B}_{t',l}$ and $\mathcal{B}_{t'}^i$ similarly and bound $\mathbb{P}(\mathcal{B}_{2l\tau,l}|\mathcal{B}_{2\tau,1}\dots\mathcal{B}_{2(l-1)\tau,l-1}) \geq 1 - \frac{4I_l}{T^2}$ by decomposing the scenarios by how the arrival and service rates are changing when considering each resource queue. $\square$

### 5.2  Regret Bounds of RNRM-UCB and RNRM-TS

The high probability bounds for mixing time of Jackson Network developed in §5.1 help bound the loss due to the nonstationarity, and thus we are ready to carry out the regret analysis of RNRM-UCB.

**Theorem 1.** *The $T$-period cumulative regret of RNRM-UCB is bounded by $\tilde{O}\left(J\sqrt{XT}\right)$.*

Note that the regret lower bound for nominal MAB (where $J = 1$) is $\Omega(\sqrt{XT})$ (see, e.g., Theorem 2.1 in Slivkins (2019)), which suggests that our regret bound is tight, if the system size $J$ is fixed.

*Proof Sketch.* The regret of the Warm-up phase is at most linear on the length of the Warm-up Phase, i.e., $O(X \log(J^{1/4}T))$. We decompose the relaxed regret in the Learning Phase into two parts. The first part is called ***loss of nonstationarity*** and it represents the revenue loss induced by the transient performance when changing the price. For batch $m$, we can derive that the expected revenue collected under the selected price $\mathbf{x}_m$ as $r(\vec{\alpha}_{\mathbf{x}_m}, \mathbf{x}_m)I_m L\tau - O(L\tau)$ with Proposition 4. The second part represents the ***loss of suboptimality***, which is induced by not offering the optimal state-independent price in certain time periods. This part can be further rewritten as $\Delta(\mathbf{x}_m)I_m L\tau$ and $\Delta(\mathbf{x}_m) = (r(\vec{\alpha}_{\tilde{\mathbf{x}}}, \tilde{\mathbf{x}}) - U_m(\tilde{\mathbf{x}})) + (U_m(\mathbf{x}_m) - L_m(\mathbf{x}_m)) + (L_m(\mathbf{x}_m) - r(\vec{\alpha}_{\mathbf{x}_m}, \mathbf{x}_m))$. We bound these three terms separately based on Proposition 2 and Lemma 2. Combine all results together, we reach an $\tilde{O}(J\sqrt{XT})$ regret bound. We provided detailed proof in Appendix E. $\square$

We use *Bayesian regret* as the performance measure of RNRM-TS, which is typically used for the Bayesian setting and is computed as the expected regret of the instance in expectation over all problem instances. Similarly, we define a relaxed Bayesian regret.

**Definition 4.** *Denoting all the events before period $t$ by $\mathcal{F}_t$, $\forall t \geq 0$, we define the relaxed Bayesian regret of policy $\pi$ by period $T$ as*

$$\overline{BR}(\pi, T) = \mathbb{E}_{\{\alpha_j(\mathbf{x})\}_{j \in \mathcal{J}} \sim priori}\left[\mathbb{E}[Y^{LP} - Y^\pi | \mathcal{F}_T]\right].$$

The relaxed Bayesian regret bound of RNRM-TS is in the same order as RNRM-UCB (Appendix E).

**Theorem 2.** *The $T$-period cumulative Bayesian regret of RNRM-TS is bounded by $\tilde{O}(J\sqrt{XT})$.*

## 6   Numerical Results

We consider $T = 2000$, $I = 2$ with capacity $c_1 = c_2 = 2$, $J = 3$ with product-resource orders $\{1, 2, 1 \rightarrow 2\}$, and $\mathcal{X} = \{[1, 2, 3], [2, 1, 3], [2, 2, 3]\}$. We set the underlying arrival rates of candidate prices as $\{[4, 2, 4], [3, 4, 3], [2.5, 3, 2.5]\}$ and use uniform service rates $[5, 4.5]$ for all prices. We adopt the blind network revenue management algorithm proposed in Besbes and Zeevi (2012) to our setting as a benchmark. This benchmark algorithm tries each price for $\theta T/X$ periods and then selects the price with the highest revenue for the rest of the periods. We set $\theta$ as 0.22, 0.1, and 0.05 in three Besbes and Zeevi (2012) benchmarks, respectively (the same as the settings in Table 1 in the original paper Besbes and Zeevi 2012). To conclude, we implement five pricing algorithms: RNRM-UCB, RNRM-TS, and three benchmark policies with different values of $\theta$.

Figure 1a shows the offered price over periods of each algorithm and Figure 1b depicts the cumulative time-average relaxed regret, i.e., $\sum_{t'=1}^{t}(Y_{t'}^{\text{LP}} - Y_{t'}^{\pi})/t$. For all algorithms (except "OPT"), the revenue is collected from the simulation where the system starts with no customers at time 0. We can observe that both RNRM-UCB and RNRM-TS identify the best price (P1) through the learning process and the time average regret shrinks to zero very quickly.

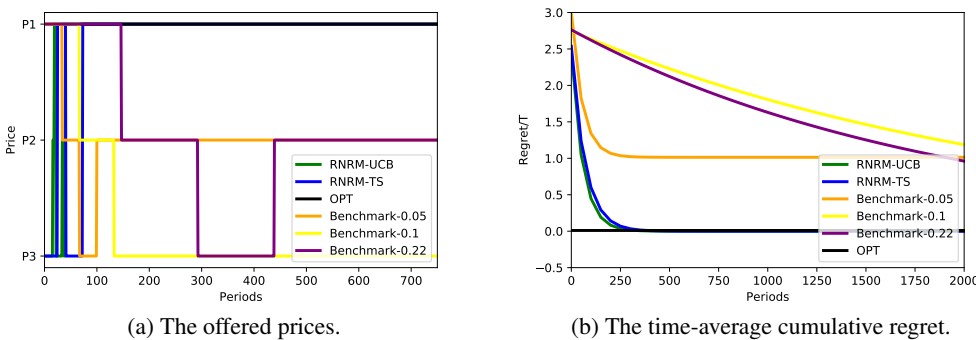

(a) The offered prices.          (b) The time-average cumulative regret.

Figure 1: Offered price and time-average cumulative regret of policies. Note that the price selections keep the same after Period 750 and therefore we only plot for Periods 1 to 750 in subfigure (a).

## 7   Conclusion and Broader Impacts

We proposed two new batch bandit algorithms for RNRM problems with multiple products and multiple reusable resources. We established optimal regret upper bounds and demonstrated superior computational results. This work solves an important practical problem and we believe that our approach will open many doors to learning more complex stochastic systems (especially in network settings). There are two immediate future research questions: (1) how we can accommodate continuous price space; and (2) how we can extend arrivals and services to general distributions. Both directions would need significant methodological innovations to be developed.

### Acknowledgments and Disclosure of Funding

The authors would like to thank anonymous reviewers for their detailed and helpful comments, which have helped improve the content and exposition of this paper. This research is partially supported by an Amazon research award and Department of Energy award #DE-SC0018018.

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
