## Appendix – "Online Learning and Pricing for Network Revenue Management with Reusable Resources" by Huiwen Jia, Cong Shi, Siqian Shen

## A Summary of Major Notation

Table 1: Summary of Major Notation for Problem Formulation

| | |
|---|---|
| $\mathcal{I}$ | the set of reusable resource (server) types |
| $\mathcal{J}$ | the set of product (service) types |
| $c_i$ | the capacity of server type $i \in \mathcal{I}$ |
| $\mu_i(\mathbf{x})$ | the service rate of server type $i$ under the price $\mathbf{x}$ |
| $\mathbf{x}$ | the price vector $\mathbf{x} = \{x_j : j \in \mathcal{J}\}$ |
| $\mathcal{X}$ | the set of candidate prices |
| $T$ | the number of periods in consideration |
| $\alpha(\mathbf{x})$ | the overall demand rate of the system under the price $\mathbf{x}$ |
| $\alpha_j(\mathbf{x})$ | the demand rate of product $j$ under the price $\mathbf{x}$ |
| $\vec{\alpha}_{\mathbf{x}}$ | the vector to demand of products under the price $\mathbf{x}$, $\vec{\alpha}_{\mathbf{x}} \in \mathbb{R}^J$ |
| $\lambda_i(\mathbf{x})$ | the demand rate of server $i$ under the price $\mathbf{x}$ |
| $A$ | the resource consumption matrix, $A_{ji} \in \mathbb{R}^{J \times I}$, where element $a_{ji} = 1$ indicates that service type $j$ requires to be processed on server type $i$ |
| $\mathcal{J}_{\text{on}}(\mathbf{x})$ | the turn-on product set under price $\mathbf{x}$, containing products with non-zero demand rates |
| $\mathcal{J}_{\text{off}}(\mathbf{x})$ | the turn-off product set under price $\mathbf{x}$, containing products with zero demand rates |
| $P$ | the server routing matrix where $p_{ii'}$ denotes the demand portion that consumes resource $i'$ straight after consuming resource $i$, $\sum_{i' \in \mathcal{I} \cup \{t\}} p_{ii'} = 1$. |
| $\mathbf{y}$ | a $J$-dimensional vector, $\mathbf{y} \in \{0, 1, \dots, \infty\}^I$, denote the number of current customers across different types of resources |
| $Y_t^\pi$ | the expected revenue under pricing policy $\pi$ during period $t$, $\forall t = 1, \dots, T$, |
| $Y^\pi$ | the cumulative expected revenue over periods $\{1, \dots, T\}$, $Y^\pi = \sum_{t=1}^T Y_t^\pi$ |
| $\tilde{\mathbf{x}}$ | the (static) optimal state-independent price, $\tilde{\mathbf{x}} = \arg\max_{\mathbf{x} \in \mathcal{X}} \vec{\alpha}_{\mathbf{x}}^T \mathbf{x}$ |
| $r(\vec{\alpha}_{\mathbf{x}}, \mathbf{x})$ | steady-state revenue rate of price $\mathbf{x}$, $r(\vec{\alpha}_{\mathbf{x}}, \mathbf{x}) = \sum_{j \in \mathcal{J}_{\text{on}}(\mathbf{x})} \mathbf{x}_j \alpha_j(\mathbf{x})$ |

Table 2: Summary of Major Notation for Algorithms and Regret Analysis

| | |
|---|---|
| $L$ | the number of layers |
| $\mathcal{I}_l$ | resources of layer $l$ |
| $M$ | the number of batches |
| $\tau$ | the value for computing the length of each batch, $\tau = (\log(T))^2$ |
| $I_m$ | batch $m$, $m = 1, \dots, M$ contains $I_m L \tau$ periods, where $I_m = 2^m$ |
| $n_j(\mathbf{x})$ | the number of arrived customers of product $j$ under price $\mathbf{x}$ |
| $\hat{d}_{jk}(\mathbf{x})$ | as the observed time interval between the arrival time of customer $k$ and that of customer $k-1$ who are requesting product $j$ under the price $\mathbf{x}$ |
| $\bar{d}_j(\mathbf{x})$ | an empirical estimate of arrival time, $\bar{d}_j(\mathbf{x}) = \sum_{k=1}^{n_j(\mathbf{x})} \hat{d}_{jk}(\mathbf{x})/n_j(\mathbf{x})$, for $j \in \mathcal{J}$ |
| $\mathbf{Rad}_t(\mathbf{x})$ | confidence radius of price $\mathbf{x}$ by the end of period $t$ |
| $U_t(\mathbf{x})$ | the upper confidence bound of expected stationary revenue rate of price $\mathbf{x}$ |
| $L_t(\mathbf{x})$ | the lower confidence bound of expected stationary revenue rate of price $\mathbf{x}$ |
| $\mathbf{x}_m$ | the offered price in batch $m$ by the algorithm |
| $Y_m(\mathbf{x})$ | the practical revenue collected in batch $m$ with price $\mathbf{x}$ |
| $\mathcal{M}_{\mathbf{x}} \subseteq \{1, \dots, M\}$ | the batches choosing price $\mathbf{x}$ |
| $T_{\mathbf{x}} = \sum_{m \in \mathcal{M}_{\mathbf{x}}} I_m \tau$ | the number of periods choosing price $\mathbf{x}$ |
| $\mathbf{Gamma}(\eta^{\alpha_j \mathbf{x}}, \beta^{\alpha_j \mathbf{x}})$ | the prior/posterior distribution of $\alpha_j(\mathbf{x})$, the demand of product $j$ under price $\mathbf{x}$ |
| $\mathcal{A}_t$ | the Jackson Network reaches the steady-state by time $t$ from empty state |
| $\mathcal{A}_{t,l}$ | all layer-l nodes reach the steady-state by time $t$ from empty state |
| $\mathcal{A}_t^i$ | the node $i$ reaches the steady-state by time $t$ from empty state |
| $\mathcal{B}_t$ | the Jackson Network reaches the steady-state by time $t$ from steady-state of old price |
| $\mathcal{B}_{t,l}$ | all layer-l nodes reach the steady-state by time $t$ from steady-state of old price |
| $\mathcal{B}_t^i$ | the node $i$ reaches the steady-state by time $t$ from steady-state of old price |
| $\mathcal{C}_{2L\tau}$ | the event that the system reaches the steady-state under the new price within the first $2L\tau$ periods |
| $\bar{\mathcal{C}}_{2L\tau}$ | the complementary event of $\mathcal{C}_{2L\tau}$ that the system does not reach the steady-state within $2L\tau$ periods |

## B Illustrative Example

We refer to this simple illustrative example several times in the rest of the paper. Consider $4$ types of resources and $4$ types of products:

- Product No.1: $(s, 1, 3, t)$, i.e., SQL Database $\rightarrow$ EC2

- Product No.2: $(s, 1, 3, 4, t)$, i.e., SQL Database $\to$ EC2 $\to$ SAP
- Product No.3: $(s, 2, 3, t)$, i.e., NoSQL Database $\to$ EC2
- Product No.4: $(s, 2, 3, 4, t)$, i.e., NoSQL Database $\to$ EC2 $\to$ SAP

Typically, customers will choose between an SQL based or a NoSQL based database service to store and preprocess the data. After running algorithms on elastic cloud compute EC2, only a subset of customers may choose to further integrate SAP applications. The network representation is given by Figure 2, and it is straightforward to check that this acyclic graph satisfies Assumption 2.

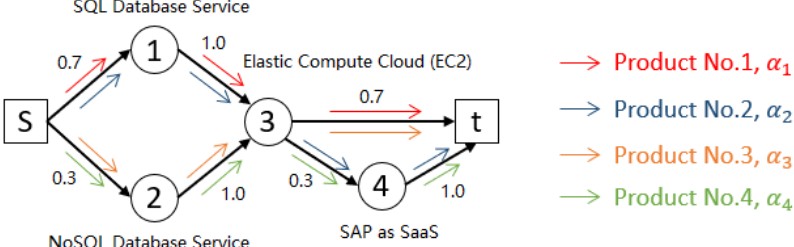

Figure 2: Servers and products in a network system.

We assign a price vector $\mathbf{x} = (x_1, x_2, x_3, x_4)$ to these four products. Note that we do not assign prices to resources. We only sell individual products (as bundles of resources). Each posted price vector induces two quantities, namely, $\alpha(\mathbf{x})$ and $P(\mathbf{x})$. Here $\alpha(\mathbf{x})$ is the total arrival rate of all products given price vector $\mathbf{x}$ while $P(\mathbf{x})$ is the so-called (Markovian) "routing matrix" that captures the demand flows for products. One can represent a product $j \in \mathcal{J}$ using an $s$-$t$ path, i.e., $(s, i_1, \ldots, i_n, t)$. Assume that Assumption 3 holds. Then we can write the demand rate for this particular product $j$ as

$$\alpha_j = \alpha(\mathbf{x}) p_{s,i_1} p_{i_1,i_2} \ldots p_{i_{n-1},i_n} p_{i_n,t},$$

where $p_{i,i'}$ are the $(i, i')$th entry of the routing probability matrix $P(\mathbf{x})$.

Note that in our setting, the mapping $\mathbf{x} \mapsto \alpha(\mathbf{x}), P(\mathbf{x})$ is unknown to the decision maker. Thus, we need to offer different price vectors $\mathbf{x}$ to learn this mapping. And we also assume this mapping is nonparametric. For the sake of illustration, we give the following linear mapping as an example.

$$\begin{aligned} \alpha(\mathbf{x}) &= 110 - \mathbf{1} \cdot \mathbf{x}, \\ p_{ii'} &= \frac{\sum_{j:a_{ji}=1, a_{ji'}=0} x_j}{\sum_{j:a_{ji}=1} x_j}. \end{aligned}$$

Suppose the price vector $\mathbf{x} = (1, 2, 2, 5)$. In this case, the total arrival rate $\alpha(\mathbf{x}) = 100$ and the routing matrix $P(\mathbf{x})$ is given by

$$P(\mathbf{x}) = \begin{cases} p_{s1} = \dfrac{x_3 + x_4}{x_1 + x_2 + x_3 + x_4} = 0.7 \\ p_{s2} = \dfrac{x_1 + x_2}{x_1 + x_2 + x_3 + x_4} = 0.3 \\ p_{3t} = \dfrac{x_2 + x_4}{x_1 + x_2 + x_3 + x_4} = 0.7 \\ p_{34} = \dfrac{x_1 + x_3}{x_1 + x_2 + x_3 + x_4} = 0.3 \\ p_{13} = p_{23} = p_{4t} = 1 \end{cases},$$

giving rise to

$$\text{Product Demand:} \begin{cases} \alpha_1 = 100 \times 0.7 \times 0.7 = 49 \\ \alpha_2 = 100 \times 0.7 \times 0.3 = 21 \\ \alpha_3 = 100 \times 0.3 \times 0.7 = 21 \\ \alpha_4 = 100 \times 0.3 \times 0.3 = 9 \end{cases}, \quad \text{Server Demand:} \begin{cases} \lambda_1 = 100 \times 0.7 = 70 \\ \lambda_2 = 100 \times 0.3 = 30 \\ \lambda_3 = 100 \times 1 = 100 \\ \lambda_4 = 100 \times 0.3 = 30 \end{cases}.$$

Note that in this example, Assumption 3 also means that out of all the customers who have chosen EC2, the proportion of choosing SAP is 0.3 and the proportion of not choosing SAP is 0.7 (regardless of database options).

# C  Technical Results in Section 3

*Proof of Proposition 1.*

We show that the decisions variables $(\pi, a, Y)$ associated with any admissible policy must satisfy the constraints of the linear program (1), and the expected revenue of this admissible policy is exactly the corresponding objective value in (1). Consider an arbitrary policy $\hat{\pi}$. We obtain the values of decision variables by the following steps. Variable $\hat{a}_{\mathbf{y}t}$ takes the value of probability that the system has $\mathbf{y}$ customers at the beginning of period $t$ under the policy $\hat{\pi}$ for $\mathbf{y} \in \{0, 1, \ldots, \infty\}^I$ and $t = 1, \ldots, T$. Therefore, constraints (1b) are immediately satisfied. Variable $\hat{\pi}^{\mathbf{x}}_{\mathbf{y}t} = 1$ if the policy $\hat{\pi}$ chooses price $\mathbf{x}$ at state $(\mathbf{y}, t)$, otherwise, $\hat{\pi}^{\mathbf{x}}_{\mathbf{y}t} = 0$, for $\mathbf{y} \in \{0, 1, \ldots, \infty\}^I$, $t = 1, \ldots, T$, $\mathbf{y} \in \mathcal{X}$. As a result, constraints (1c) hold because in the left-hand side we have one variable equal to $1$ and others all zero. Variable $\hat{Y}_{\mathbf{y}t}$ takes the value of expected revenue collected before the end of period $T$ from the customers who arrive during period $t$, when the system has $\mathbf{y}$ customers at the beginning of period $t$, for $\mathbf{y} \in \{0, 1, \ldots, \infty\}^I$. Consider arbitrary $\mathbf{y}$ and $t$, we denote the price chosen at state $(\mathbf{y}, t)$ by the policy $\hat{\pi}$ as $\hat{\mathbf{x}}$. Then, by plugging in $\hat{\pi}^{\hat{\mathbf{x}}}_{\mathbf{y}t} = 1$, in the right-hand side of constraints (1d), we have $\vec{\alpha}^T_{\mathbf{x}}\mathbf{x}$, where $\vec{\alpha}^T_{\mathbf{x}}$ denotes the average number of arrived customers for consuming products during period $t$ and $\hat{\mathbf{x}}$ is the revenue vector of these products. Therefore, $\vec{\alpha}^T_{\mathbf{x}}\mathbf{x}$ denotes the expected revenue collected from customers who arrive during period $t$ (assuming all arrived customers are served successfully). Consequently, we can conclude that the right-hand side of (1d) is as large as the left-hand side, because the right-hand side may also compute the revenue collected after the end of period $T$. Hence, constraints (1d) are also satisfied. By the value assignment steps of $\hat{a}_{\mathbf{y}t}$ and $\hat{Y}_{\mathbf{y}t}$, for $\mathbf{y} \in \{0, 1, \ldots, \infty\}^I$, $t = 1, \ldots, T$, one has that the corresponding LP objective value at solution $(\hat{\pi}, \hat{a}, \hat{J})$ is $Y^{\hat{\pi}}$. Therefore, we have $Y^{LP} \geq Y^{\pi}$, for any admissible policy $\pi$, and thus also for the optimal state-dependent policy $\pi^*$.

By defining $\tilde{\mathbf{x}} = \text{argmax}_{\mathbf{x} \in \mathcal{X}} \vec{\alpha}^T_{\mathbf{x}}\mathbf{x}$, we find an optimal solution $(\tilde{\pi}, \tilde{a}, \tilde{Y})$ to the LP in (1) as follows.

$$\tilde{\pi}^{\tilde{\mathbf{x}}}_{\mathbf{y}t} = 1, \quad \forall \mathbf{y} \in \{0, 1, \ldots, \infty\}^I, \; \forall t = 1, \ldots, T \tag{3a}$$

$$\tilde{\pi}^p_{\mathbf{y}t} = 0, \quad \forall \mathbf{y} \in \{0, 1, \ldots, \infty\}^I, \; \forall t = 1, \ldots, T, \; \forall \mathbf{x} \in \mathcal{X} \backslash \{\tilde{\mathbf{x}}\} \tag{3b}$$

$$\tilde{Y}_{\mathbf{y}_0 t} = \vec{\alpha}^T_{\tilde{\mathbf{x}}}\tilde{\mathbf{x}} \quad \forall t = 1, \ldots, T \tag{3c}$$

$$\tilde{Y}_{\mathbf{y}t} = 0 \quad \forall \mathbf{y} \in \{0, 1, \ldots, \infty\}^I \backslash \{\mathbf{y}_0\}, \; \forall t = 1, \ldots, T \tag{3d}$$

$$\tilde{a}_{\mathbf{y}_0 t} = 1 \quad \forall t = 1, \ldots, T \tag{3e}$$

$$\tilde{a}_{\mathbf{y}t} = 0 \quad \forall \mathbf{y} \in \{0, 1, \ldots, \infty\}^I \backslash \{\mathbf{y}_0\}, \; \forall t = 1, \ldots, T, \tag{3f}$$

where $\mathbf{y}_0 = [0]^J$. Then, we can compute $Y^{LP} = T \cdot \vec{\alpha}^T_{\tilde{\mathbf{x}}}\tilde{\mathbf{x}}$. Note that $\vec{\alpha}^T_{\mathbf{x}}\mathbf{x}$ is the stationary revenue rate of price $\mathbf{x}$, denoted by $r(\vec{\alpha}_{\mathbf{x}}, \mathbf{x})$.

**Q.E.D.**

**Discussion over state variable definitions.** In LP (1), we use $(\mathbf{y}, t)$ as the state variable, where $\mathbf{y}$ reflects the number of customers who are consuming or in the waiting queue of each type of resource. With Proposition 1, we know that $Y^{LP} = T \cdot \vec{\alpha}^T_{\tilde{\mathbf{x}}}\tilde{\mathbf{x}}$ is a valid upper bound of the state-dependent optimal policy with $(\mathbf{y}, t)$ the state variable, i.e., $Y^* \leq Y^{LP}$. One may consider a different (and more refined) definition for the state variable. Consider $(W, t)$ as the new state variable with $W \in \mathbb{R}^{I \times J}$, where $W_{ij}$ represents the number of customers who are requesting product $j$ and currently consuming or in the waiting queue of resource $i$. We denote the optimal state-dependent policy under this new state variable definition by $\pi^{**}$ and the corresponding expected revenue by $Y^{**}$. Then we have two results. First, it is evident that the optimal state-dependent policy under the original state variable definition is a feasible policy for the new state variable definition and thus $Y^{**} \geq Y^*$. Second, we still have $Y^{**} \leq Y^{LP}$. To prove this, we can replace the state variables in decision variables in LP (1) to obtain a new linear program, denoted by LP2. Then following the same analysis as in Proposition 1, we can readily show that the optimal objective value of LP2 is a valid upper bound of $Y^{**}$ and

$Y^{\text{LP2}} = Y^{\text{LP}} = T \cdot \vec{\alpha}_{\tilde{\mathbf{x}}}^T \tilde{\mathbf{x}}$. Therefore, we have $Y^{**} \leq Y^{\text{LP2}} = Y^{\text{LP}}$. In fact, under any valid state variable definition, $Y^{\text{LP}}$ provides an upper bound of the state-dependent optimal policy.

## D Technical Results in Section 4.

*Proof of Proposition 2.*

The proof is based on the concentration bound for sub-exponential random variables. For product $j \in \mathcal{J}_{\text{off}}$, their demand is off, i.e., $\alpha_j(\mathbf{x}) = 0$, $j \in \mathcal{J}_{\text{off}}(\mathbf{x})$. In this proof, we only consider one price and thus we use $\alpha_j$ to represent $\alpha_j(\mathbf{x})$ for notation simplicity. We can rewrite the right-hand-side as:

$$
\begin{aligned}
&\mathbb{P}\left( \left| \vec{\alpha}_{\mathbf{x}}^T \mathbf{x} - \sum_{j \in \mathcal{J}_{\text{on}}(\mathbf{x})} \frac{x_j}{\bar{d}_j(\mathbf{x})} \right| \leq \sum_{j \in \mathcal{J}_{\text{on}}(\mathbf{x})} \frac{x_j \sqrt{32 \log(J^{\frac{1}{4}} T)}}{\bar{d}_j(\mathbf{x}) \sqrt{n_j(\mathbf{x})}} \right) \\
=&\mathbb{P}\left( \left| \sum_{j \in \mathcal{J}_{\text{on}}(\mathbf{x})} \left( \alpha_j x_j - \frac{x_j}{\bar{d}_j(\mathbf{x})} \right) \right| \leq \sum_{j \in \mathcal{J}_{\text{on}}(\mathbf{x})} \frac{x_j \sqrt{32 \log(J^{\frac{1}{4}} T)}}{\bar{d}_j(\mathbf{x}) \sqrt{n_j(\mathbf{x})}} \right) \\
\geq&\mathbb{P}\left( \sum_{j \in \mathcal{J}_{\text{on}}(\mathbf{x})} \left| \alpha_j x_j - \frac{x_j}{\bar{d}_j(\mathbf{x})} \right| \leq \sum_{j \in \mathcal{J}_{\text{on}}(\mathbf{x})} \frac{x_j \sqrt{32 \log(J^{\frac{1}{4}} T)}}{\bar{d}_j(\mathbf{x}) \sqrt{n_j(\mathbf{x})}} \right) \\
\geq& \prod_{j \in \mathcal{J}_{\text{on}}(\mathbf{x})} \mathbb{P}\left( \left| \alpha_j x_j - \frac{x_j}{\bar{d}_j(\mathbf{x})} \right| \leq \frac{x_j \sqrt{32 \log(J^{\frac{1}{4}} T)}}{\bar{d}_j(\mathbf{x}) \sqrt{n_j(\mathbf{x})}} \right).
\end{aligned}
\tag{4}
$$

We can derive that for one product $j \in \mathcal{J}_{\text{on}}(\mathbf{x})$:

$$
\mathbb{P}\left( \left| \alpha_j x_j - \frac{x_j}{\bar{d}_j(\mathbf{x})} \right| \leq \frac{x_j \sqrt{32 \log(J^{\frac{1}{4}} T)}}{\bar{d}_j(\mathbf{x}) \sqrt{n_j(\mathbf{x})}} \right) = \mathbb{P}\left( \left| \alpha_j - \frac{1}{\bar{d}_j(\mathbf{x})} \right| \leq \frac{\sqrt{32 \log(J^{\frac{1}{4}} T)}}{\bar{d}_j(\mathbf{x}) \sqrt{n_j(\mathbf{x})}} \right) \tag{5a}
$$

$$
= \mathbb{P}\left( \left| \frac{\alpha_j - \frac{1}{\bar{d}_j(\mathbf{x})}}{\alpha_j \frac{1}{\bar{d}_j(\mathbf{x})}} \right| \leq \frac{\sqrt{32 \log(J^{\frac{1}{4}} T)}}{\alpha_j \frac{1}{\bar{d}_j(\mathbf{x})} \bar{d}_j(\mathbf{x}) \sqrt{n_j(\mathbf{x})}} \right) = \mathbb{P}\left( \left| \bar{d}_j(\mathbf{x}) - \frac{1}{\alpha_j} \right| \leq \frac{\sqrt{32 \log(J^{\frac{1}{4}} T)}}{\alpha_j \sqrt{n_j(\mathbf{x})}} \right). \tag{5b}
$$

The exponential arrival time intervals are also sub-exponential random variables with parameters $(\frac{4}{\alpha_j^2}, \frac{2}{\alpha_j})$, denoted by $\text{SE}(\frac{4}{\alpha_j^2}, \frac{2}{\alpha_j})$, see Lemma 1 in the appendix. Consider the concentration inequality for i.i.d. exponential variables $\xi_i$, $i = 1, \ldots n$ with mean $1/\lambda$:

$$
\mathbb{P}\left[ \left| \frac{1}{\lambda} - \frac{1}{n} \sum_{i=1}^n \xi_i \right| \geq t \right] \leq \begin{cases} 2 \exp\left( -\frac{n t^2}{2 \frac{4}{\lambda^2}} \right) & \text{if } 0 \leq t \leq \frac{2}{\lambda} \\ 2 \exp\left( -\frac{n t}{2 \frac{2}{\lambda}} \right) & \text{if } t > \frac{2}{\lambda} \end{cases}. \tag{6}
$$

By the assumption $n_j(\mathbf{x}) \geq 8 \log(J^{\frac{1}{4}} T)$, we can compute

$$
\left( \frac{\sqrt{32 \log(J^{\frac{1}{4}} T)}}{\alpha_j \sqrt{n_m(\mathbf{x})}} \right)^2 = \frac{32 \log(J^{\frac{1}{4}} T)}{\alpha_j^2 n_j(\mathbf{x})} \leq \frac{4}{\alpha_j^2},
$$

and thus we can apply the first inequality in (6) to (5). Therefore, we can derive

$$
\mathbb{P}\left( \left| \bar{d}_j(\mathbf{x}) - \frac{1}{\alpha_j} \right| \leq \frac{\sqrt{32 \log(J^{\frac{1}{4}} T)}}{\alpha_j \sqrt{n_j(\mathbf{x})}} \right) \geq 1 - 2 \exp\left( \frac{-n_j(\mathbf{x}) \frac{32 \log(J^{\frac{1}{4}} T)}{\alpha_j^2 n_j(\mathbf{x})}}{2 \frac{4}{\alpha_j^2}} \right) \geq 1 - \frac{2}{J \cdot T^4} \tag{7}
$$

Then, we can compute a lower bound of (4) as

$$\left(1 - \frac{2}{J \cdot T^4}\right)^{|\mathcal{J}_{\mathrm{on}}(\mathbf{x})|} \geq \left(1 - \frac{2}{J \cdot T^4}\right)^J \geq 1 - \frac{2J}{J \cdot T^4} = 1 - \frac{2}{T^4}.$$

**Q.E.D.**

**Lemma 1.** *(Sub-exponential property, Lemma 3 in Jia et al. 2020) If a random variable $X$ follows an exponential distribution with mean $1/\lambda$, then the random variable $X - \frac{1}{\lambda}$ is $(\frac{4}{\lambda^2}, \frac{2}{\lambda})$-sub-exponential.*

## E  Technical Results in Section 5

*Proof of Proposition 3.*

In this proof, we analyze the coupling time of nodes layer by layer. We define events as follows (all starting from empty state).

$$\mathcal{A}_t = \{\text{the Jackson Network reaches the steady-state by time } t\},$$
$$\mathcal{A}_{t,l} = \{\text{all layer-l nodes reach the steady-state by time } t\},$$
$$\mathcal{A}_t^i = \{\text{the node } i \text{ reaches the steady-state by time } t\}.$$

First, we discuss the coupling analysis of an M/M/c queue, which is the basis for completing this proof. With the assumption $\log > 4$, the probability that an M/M/c queue reaches the steady-state from empty state after $(\log T)^2$ periods of time is bounded by $1 - 2/T^2$, see Lemma 3 (Proposition 4 in Jia et al. (2020)). These results are derived following the law of total probability, the first-order stochastic dominance between busy times of an M/M/c queue, and the concentration inequality for independent samples. We denote node $i$ is a prerequisite node of $i'$, $i < i'$ if there exists a product requires to use nodes $i$ and $i'$. Thus, it is easy to show that the prerequisite nodes of a Layer-$l$ node must have smaller layer indices.

In our Jackson Network, node $i'$ reaches its steady-state indicates that the arrival process of node $i'$ is a stable Poisson process, and therefore it requires that all the prerequisite nodes also reach the steady-state. In this proof, we analyze the nodes layer by layer. By treating the nodes in the same layer together, we can derive that

$$
\begin{aligned}
\mathbb{P}(\mathcal{A}_{L\tau}) &= \mathbb{P}(\mathcal{A}_{L\tau,1}\mathcal{A}_{L\tau,2}\mathcal{A}_{L\tau,3}\ldots\mathcal{A}_{L\tau,L}) \\
&\geq \mathbb{P}(\mathcal{A}_{\tau,1}\mathcal{A}_{2\tau,2}\mathcal{A}_{3\tau,3}\ldots\mathcal{A}_{L\tau,L}) \\
&= \mathbb{P}(\mathcal{A}_{\tau,1})\mathbb{P}(\mathcal{A}_{2\tau,2}\mathcal{A}_{3\tau,3}\ldots\mathcal{A}_{L\tau,L}|\mathcal{A}_{\tau,1}) \\
&= \mathbb{P}(\mathcal{A}_{\tau,1})\mathbb{P}(\mathcal{A}_{2\tau,2}|\mathcal{A}_{\tau,1})\mathbb{P}(\mathcal{A}_{3\tau,3}\ldots\mathcal{A}_{L\tau,L}|\mathcal{A}_{\tau,1}\mathcal{A}_{2\tau,2}) \\
&= \mathbb{P}(\mathcal{A}_{\tau,1})\prod_{l=2}^{L}\mathbb{P}(\mathcal{A}_{l\tau,l}|\mathcal{A}_{\tau,1}\ldots\mathcal{A}_{(l-1)\tau,l-1}).
\end{aligned}
\tag{8}
$$

In short, we derive a lower bound of this mixing probability by requiring the Layer-$l$ nodes to reach the steady-state within the first $L \times \tau$ periods, conditional on that Layer-$l'$ nodes also reach the steady-state within the first $l' \times \tau$ periods for $l' < l$, or equivalently, all prerequisite nodes of Layer-$l$ nodes reach the steady-state within $(l - 1) \times \tau$ periods.

**Layer-1 Nodes.** In the first $\tau$ periods, by applying the probability bound for an M/M/c queue, we can easily conclude that all Layer-1 nodes (the correspondent M/M/c queues) reach the steady-state with probability $1 - \frac{2}{T^3}$. Recall $\mathcal{I}_1$ is the collection of all Layer-1 nodes and $|\mathcal{I}_1|$ is the cardinality. Therefore we can compute that the probability that all Layer-1 nodes reach the steady-state by first $\tau$ periods is

$$\mathbb{P}(\mathcal{A}_{\tau,1}) = \left(1 - \frac{2}{T^3}\right)^{|\mathcal{I}_1|} \geq 1 - \frac{2|\mathcal{I}_1|}{T^2},\tag{9}$$

because the queueing evolution of all Layer-1 nodes are independent of each other and thus we can simply multiply the probability together, in the first equality in the above expression.

**Layer-2 Nodes.** We then analyze the Layer-2 nodes. Consider one arbitrary layer-2 node, denoted by $i$ without loss of generality. We then focus on the corresponding queue of this node $i$ conditional on

that all Layer-1 nodes have reached steady-state. We firstly show a virtual queue can reach steady-state within the next $\tau$ periods with probability at least $1 - \frac{2}{T^2}$. Then we show that an alternative queue of the actual queue takes a shorter time to reach steady-state and thus the actual queue can also reach steady-state within the next $\tau$ periods with probability at least $1 - \frac{2}{T^2}$.

- *Virtual queue.* We consider a virtual queue that rejects all customers who arrived before $\tau$ periods. Then by Lemma 3, this virtual queue reaches the steady-state with probability $1 - \frac{2}{T^2}$ after the next $\tau$ periods, i.e., after the first $2\tau$ periods.

- *Alternative queue.* Denote the probability that the number of customers in Node $i$ at the end of period $\tau$ is less than $N$ by $\mathbb{P}_i(N)$, including the customers both using the server or in the waiting queue of node $i$. The actual queue accepts customers during the first $\tau$ periods with a changing rate from an initial rate to the stable arrival rate $\lambda_i$ (conditional on that all Layer-1 nodes reach steady-state and thus provides stable Poisson arrivals for nodes with high layer indices). Denote the demand for all products to the Jackson Network by $\alpha$. The stable arrival rate $\lambda_i = \alpha p_{si} + \alpha \sum_{i':(i',i \in E)} p_{si'} p_{i'i}$, where $\alpha p_{si}$ is the initial rate, i.e., the demand rate of products that request node $i$ as the first service, and $\alpha p_{si'} p_{i'i}$ is the stable demand flowing from prerequisite node $i'$ to node $i$. Consider an alternative queue that accepts customers at the stable rate $\lambda_i$. If the queue starts from time $0$, then the probability that this virtual queue has customers with a size less than $N$ at the end of period $\tau$ is smaller than $\mathbb{P}_i(N)$. This is because this alternative queue accepts more customers while the service times are all following the same distribution. Then we postpone the starting point of this alternative queue to a time stamp between $0$ and $\tau$ such that the probability that this alternative queue has customers with a size less than $N$ at the end of period $\tau$ is the same as $\mathbb{P}_i(N)$. Note that this timestamps exists, which can be easily proved by the memoryless property of the exponential service time assumptions. Then we have the following two results. First, this alternative queue has the same evolution path as the actual queue. Second, because this alternative queue starts accepting customers earlier than time $\tau$, and thus it will reach steady-state earlier than the virtual queue.

We provide an illustration of the actual, alternative, and virtual queues in Figure 3.

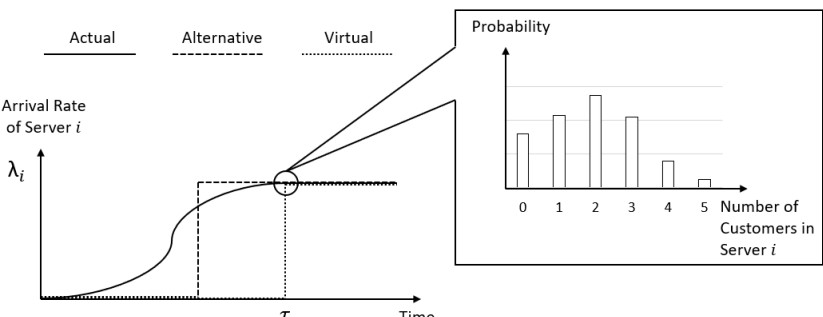

Figure 3: The left part shows the arrival rates of the actual, alternative, and virtual queues (for a Layer-2 node $i$ when the Jackson network starting from empty state). The right part shows the probability that the queue has a different number of customers at time $\tau$. These probabilities are the same for the alternative and the actual queues and thus they are equivalent systems after time $\tau$.

To conclude, the actual queue can reach the steady-state with a higher probability after the first $2\tau$ periods. Similarly, the queueing evolution of all Layer-2 nodes are independent of each other and thus we can simply multiply the probability together. Therefore, we have that for all Layer-2 nodes

$$\mathbb{P}(\mathcal{A}_{2\tau,2}|\mathcal{A}_{\tau,1}) = \left(1 - \frac{2}{T^2}\right)^{|\mathcal{I}_2|} \geq 1 - \frac{2|\mathcal{I}_2|}{T^2}. \tag{10}$$

**Layers with Higher Indices.** Then we apply the same analysis steps of the Layer-2 nodes to nodes in layers with higher indices. Conditional on the events that all nodes in layers with lower indices

have reached the steady-state by the end of $(l-1)\tau$ periods, we can compute that all nodes of layer $l$ can reach steady-state in the next $\tau$ period with a probability:

$$\mathbb{P}(\mathcal{A}_{l\tau,l}|\mathcal{A}_{\tau,1}\ldots\mathcal{A}_{(l-1)\tau,l-1}) \geq 1 - \frac{2I_l}{T^2}. \tag{11}$$

With the results in (9), (10), and (11), we are ready to bound the probability in (8):

$$\mathbb{P}(\mathcal{A}_{L\tau}) \geq \prod_{l=1}^{L}\left(1 - \frac{2|\mathcal{I}_l|}{T^2}\right) \geq 1 - \frac{2I}{T^2}.$$

We refer to Figure 4 for a direct representation of the above proof sketch, which shows the mixing time analysis of an illustrative example.

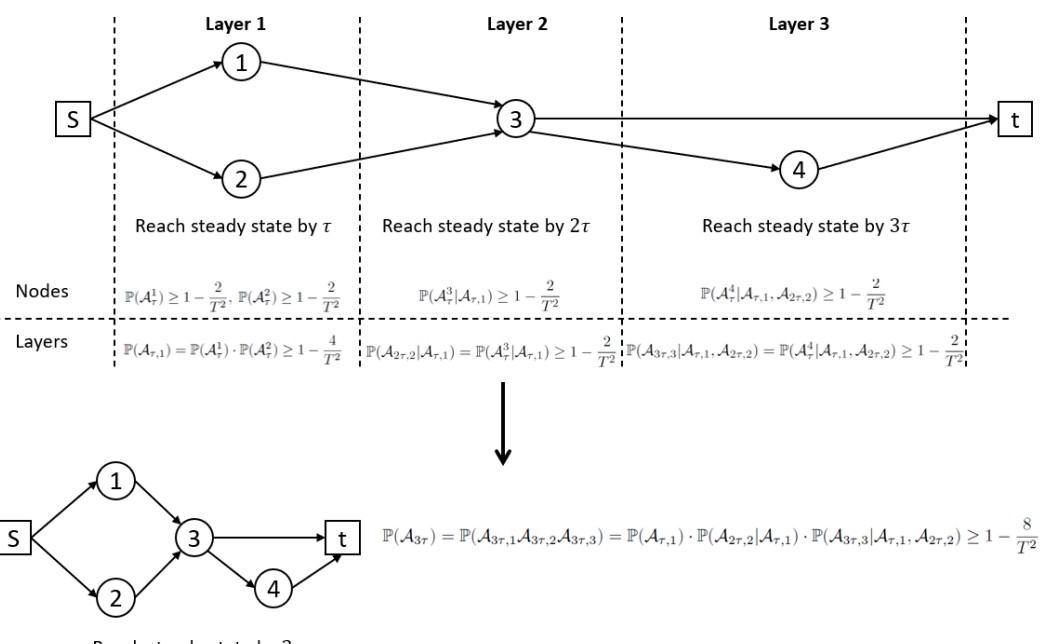

Figure 4: The mixing time analysis for an illustrative example starting from zero state.

**Q.E.D.**

*Proof of Proposition 4.*

Similar to the proof of Proposition 4, we analyze the mixing time of nodes layer by layer. Before conducting the proof, we introduce an existing work for an M/M/c queue mixing time. We similarly define the following events when the Jackson Network starts to operate under a new price starting from the steady-state of the old price:

$$\mathcal{B}_t = \{\text{the Jackson Network reaches the steady-state by time } t\},$$

$$\mathcal{B}_{t,l} = \{\text{all layer-l nodes reach the steady-state by time } t\},$$

$$\mathcal{B}_t^i = \{\text{the node } i \text{ reaches the steady-state by time } t\}.$$

Lemma 4 says that for one M/M/c queue where Assumption 5 holds, then the probability that the queue reaches a new steady-state from an old steady-state within $2\tau$ time periods after the price change is bounded by $1 - 4/T^2$. To derive this result, the authors decompose possible price changes into the following two cases. The first is when the arrival rate changes from a higher value to a lower value and the second is vice versa. For each case, the authors further consider two sub-cases: the unit service rate increase or decrease.

Consider the Jackson Network system in our problem. We can bound the probability by:

$$\mathbb{P}(\mathcal{B}_{2L\tau}) \geq \mathbb{P}(\mathcal{B}_{2\tau,1}) \prod_{l=2}^{L} \mathbb{P}(\mathcal{B}_{2l\tau,l} | \mathcal{B}_{2\tau,1} \ldots \mathcal{B}_{2(l-1)\tau,l-1}). \tag{12}$$

The above inequality is derived following the same steps in (8) and we neglect the details here.

Assumption 5 allows us to apply Lemma 4 when we analyze one specific node, i.e., one specific M/M/c queue. The structure of the following analysis is similar to the structure of the proof of Proposition 3, we first analyze Layer-1 nodes and then the nodes with higher layer indices. For nodes with higher layer indices, we apply the aforementioned proposition to a virtual queue, and then show this virtual queue takes longer than an alternative system with the actual queue.

**Layer-1 Nodes.** In the first $2\tau$ periods, by applying the probability bound for an M/M/c queue with price change (Lemma 4), we can easily conclude that all Layer-1 nodes (the correspondent M/M/c queues) reach the steady-state with probability $1 - \frac{4}{T^2}$. Consider the probabilistic independency of the coupling process among Layer-1 nodes, the probability that all Layer-1 nodes reach the steady-state by the first $2\tau$ periods is

$$\mathbb{P}(\mathcal{B}_{2\tau,1}) = \left(1 - \frac{4}{T^2}\right)^{|\mathcal{I}_1|} \geq 1 - \frac{4|\mathcal{I}_1|}{T^2}. \tag{13}$$

**Layer-2 Nodes and Nodes with Higher Layer Indices.** We then analyze nodes with a layer index 2 or higher. Consider one arbitrary Layer-l node, denoted by $i$ without loss of generality. We then focus on the corresponding queue of this node $i$ conditional on that all prerequisites nodes have reached steady-state. We firstly show a virtual queue can reach steady-state within the next $2\tau$ periods with probability at least $1 - \frac{4}{T^2}$. Then we show that an alternative queue of the actual queue takes a shorter time to reach steady-state and thus the actual queue can also reach steady-state within the next $2\tau$ periods with probability at least $1 - \frac{4}{T^2}$.

- *Virtual queue.* We consider a virtual queue that accepts customers under the old queueing parameters in the first $2(l-1)\tau$ periods and then accepts customers under the new queueing parameter in the next $2\tau$ periods. With Assumption 5, we can directly apply the results for one M/M/c queue to reach steady-state starting from a steady-state and conclude that this virtual queue reaches the steady-state with probability $1 - \frac{4}{T^2}$ after the next $2\tau$ periods, i.e., after the first $2l\tau$ periods.

- *Alternative queue.* Denote the probability that in Node $i$ at the end of $2(l-1)\tau$ periods, the number of old customers is less than $N_o$ and the number of new customers is less than $N_n$ by $\mathbb{P}_i(N_o, N_n)$, including the customers both using the server or in the waiting queue of node $i$. Consider an alternative queue that starts accepting new customers under the new queueing system parameter from time 0 and denote the corresponding probability for having less than $N_o$ number of old customers and less than $N_n$ new customers at the end of $2(l-1)\tau$ periods by $\mathbb{P}'_i(N_o, N_n)$. By the analysis in Proposition 3, we know there is a mismatch between $\mathbb{P}_i(N_o, N_n)$. Then we postpone the starting point of this alternative queue to a time stamp between 0 and $2(L-1)\tau$ such that $\mathbb{P}'_i(N_o, N_n) = \mathbb{P}_i(N_o, N_n)$. Note that these timestamps always exist, which can be easily proved by the memoryless property of the exponential service time assumptions because we can always find this stamp backward from any customer state at time $2(l-1)\tau$. Then we have the following two results. First, this alternative queue has the same evolution path as the actual queue. Second, because this alternative queue starts accepting new customers earlier than time $2(l-1)\tau$, and thus it will reach steady-state earlier than the virtual queue.

To conclude, the actual queue can reach the steady-state with a higher probability after the first $2\tau$ periods. Therefore, conditional on the events that all nodes in layers with lower indices have reached the steady-state by the end of $2(l-1)\tau$ periods, we can compute that all nodes of layer $l$ can reach steady-state in the next $2\tau$ period with a probability:

$$\mathbb{P}(\mathcal{B}_{2l\tau,l} | \mathcal{B}_{2\tau,1} \ldots \mathcal{B}_{2(l-1)\tau,l-1}) \geq 1 - \frac{4I_l}{T^2}. \tag{14}$$

With the results in (13) and (14), we are ready to bound the probability in (12):

$$\mathbb{P}(\mathcal{B}_{2L\tau}) \geq \prod_{l=1}^{L} \left( 1 - \frac{4|\mathcal{I}_l|}{T^2} \right) \geq 1 - \frac{4I}{T^2}.$$

**Q.E.D.**

*Proof of Theorem 1.*

We bound the relaxed regret during the Warm-up and Learning Phases separately. We further decompose the relaxed regret during the Learning Phase into two parts. The first part is the regret of suboptimality, which is induced by not offering the state-independent optimal price in certain time periods. We bound the loss due to suboptimality with the help of Proposition 2. The second part is the regret of nonstationarity, which represents the revenue loss induced by changing offered prices. We bound the loss due to nonstationarity with the help of Propositions 3 and 4. The structure of this proof is similar to that of Theorem 1 in Jia et al. (2020), where the authors consider a single reusable product setting.

**Regret of the Warm-up Phase.** The difference between $Y_{\text{Warm-up}}^{\text{LP}}$ and $Y_{\text{Warm-up}}^{\text{RNRM-UCB}}$ is at most linear on the length of the Warm-up Phase, which is $O(X \log(J^{1/4}T))$. Therefore we can directly have

$$\overline{\text{Regret}}_{\text{Warm-up}}(\text{RNRM-UCB}, T) = O(X \log(J^{1/4}T)). \tag{15}$$

**Regret of nonstationarity during the Learning Phase.** We denote the offered price of batch $m$ in RNRM-UCB as $\mathbf{x}_m$. Therefore, the expected revenue collected in batch $m$ is

$$Y_m^{\text{RNRM-UCB}}(\mathbf{x}_m) = Y_{m,0\sim 2L\tau}^{\text{RNRM-UCB}}(\mathbf{x}_m) + Y_{m,2L\tau\sim I_m L\tau}^{\text{RNRM-UCB}}(\mathbf{x}_m),$$

where $Y_{m,0\sim 2L\tau}^{\text{RNRM-UCB}}(\mathbf{x}_m)$ denotes the expected revenue collected in the first $2L\tau$ periods and $Y_{m,2L\tau\sim I_m L\tau}^{\text{RNRM-UCB}}(\mathbf{x}_m)$ denotes the expected revenue collected from $(2L\tau + 1)$th period to the end of batch $m$. We define event $\mathcal{C}_{2L\tau}$ as that the network service system reaches the steady-state under the new price within the first $2L\tau$ periods and define $\bar{\mathcal{C}}_{2L\tau}$ as the complementary event of $\mathcal{C}_{2L\tau}$. By Propositions 3 and 4, we have $\mathbb{P}[\mathcal{C}_{2L\tau}] \geq 1 - 4I/T^2$. Consequently, we define $Y_{m,\mathcal{C}_{2L\tau}}^{\text{RNRM-UCB}}(\mathbf{x}_m)$ to be the expected value of $Y_{m,2L\tau\sim I_m L\tau}^{\text{RNRM-UCB}}(\mathbf{x}_m)$ conditional on $\mathcal{C}_{2L\tau}$ and $Y_{m,\bar{\mathcal{C}}_{2L\tau}}^{\text{RNRM-UCB}}(\mathbf{x}_m)$ similarly. As a result, we can express the expected revenue during batch $m$ as:

$$
\begin{aligned}
Y_m^{\text{RNRM-UCB}}(\mathbf{x}_m) &= Y_{m,0\sim 2L\tau}^{\text{RNRM-UCB}}(\mathbf{x}_m) + Y_{m,\mathcal{C}_{2L\tau}}^{\text{RNRM-UCB}}(\mathbf{x}_m)\mathbb{P}(\mathcal{C}_{2L\tau}) + Y_{m,\bar{\mathcal{C}}_{2L\tau}}^{\text{RNRM-UCB}}(\mathbf{x}_m)\big(1 - \mathbb{P}(\mathcal{C}_{2L\tau})\big) \\
&\geq O(L\tau) + \big(\vec{\alpha}_{\mathbf{x}_m}^T \mathbf{x}_m\big)(I_m L\tau - 2L\tau)\left(1 - \frac{4I}{T^2}\right) + O\left(\frac{(I_m L\tau - 2L\tau)I}{T^2}\right) \\
&= \big(\vec{\alpha}_{\mathbf{x}_m}^T \mathbf{x}_m\big)I_m L\tau - O(L\tau) \\
&= r(\vec{\alpha}_{\mathbf{x}_m}, \mathbf{x}_m)I_m L\tau - O(L\tau).
\end{aligned}
\tag{16}
$$

**Regret of suboptimality during the Learning Phase.** Recall the upper and lower confidence bounds of the revenue rate defined in Definition 2. It is clear that

$$U_m(\mathbf{x}) - L_m(\mathbf{x}) = 2\mathbf{Rad}_m(\mathbf{x}) = 2 \sum_{j \in \mathcal{J}_{\text{on}}(\mathbf{x})} \frac{x_j \sqrt{32 \log(J^{\frac{1}{4}}T)}}{\bar{d}_j(\mathbf{x})\sqrt{n_j(\mathbf{x})}}.$$

For batch $m$, we can compute the difference in the revenue rates between the optimal price and selected price $\mathbf{x}_m$ as

$$
\begin{aligned}
\Delta(\mathbf{x}_m) &= r(\vec{\alpha}_{\tilde{\mathbf{x}}}, \tilde{\mathbf{x}}) - r(\vec{\alpha}_{\mathbf{x}_m}, \mathbf{x}_m) & \text{(17a)} \\
&= \big(r(\vec{\alpha}_{\tilde{\mathbf{x}}}, \tilde{\mathbf{x}}) - U_m(\mathbf{x}_m)\big) + \big(U_m(\mathbf{x}_m) - L_m(\mathbf{x}_m)\big) + \big(L_m(\mathbf{x}_m) - r(\vec{\alpha}_{\mathbf{x}_m}, \mathbf{x}_m)\big) & \text{(17b)} \\
&\leq \big(r(\vec{\alpha}_{\tilde{\mathbf{x}}}, \tilde{\mathbf{x}}) - U_m(\tilde{\mathbf{x}})\big) + \big(U_m(\mathbf{x}_m) - L_m(\mathbf{x}_m)\big) + \big(L_m(\mathbf{x}_m) - r(\vec{\alpha}_{\mathbf{x}_m}, \mathbf{x}_m)\big), & \text{(17c)}
\end{aligned}
$$

where the last inequality holds because $U_m(\mathbf{x}_m) \geq U_m(\tilde{\mathbf{x}})$ by the algorithm construction.

Then the cumulative regret during the Learning Phase is

$$\overline{\text{Regret}}_{\text{Learning}}(\text{RNRM-UCB}, T) = Y_{\text{Learning}}^{\text{LP}} - Y_{\text{Learning}}^{\text{RNRM-UCB}} \tag{18a}$$

$$= \sum_{m=1}^{M} \left( r\left(\alpha_{\tilde{\mathbf{x}}}, \tilde{\mathbf{x}}\right) I_m L\tau - Y_m^{\text{RNRM-UCB}}\left(\mathbf{x}_m\right) \right) \tag{18b}$$

$$= \sum_{m=1}^{M} \mathbb{E}\left[ \Delta(\mathbf{x}_m) I_m L\tau + O(L\tau) \right]. \tag{18c}$$

By (17), we have

$$\sum_{m=1}^{M} \mathbb{E}\left[ \Delta(\mathbf{x}_m) I_m L\tau \right] = \sum_{m=1}^{M} \mathbb{E}\left[ \left( r(\vec{\alpha}_{\tilde{\mathbf{x}}}, \tilde{\mathbf{x}}) - U_m(\tilde{\mathbf{x}}) \right) \right] I_m L\tau + \sum_{m=1}^{M} \mathbb{E}\left[ \left( U_m(\mathbf{x}_m) - L_m(\mathbf{x}_m) \right) \right] I_m L\tau$$

$$+ \sum_{m=1}^{M} \mathbb{E}\left[ \left( L_m(\mathbf{x}_m) - r(\vec{\alpha}_{\mathbf{x}_m}, \mathbf{x}_m) \right) \right] I_m L\tau. \tag{19}$$

We first analyze the middle term of (19). Let $\mathcal{M}_{\mathbf{x}}$ denote the batches choosing price $\mathbf{x}$ and $T_{\mathbf{x}}$ the number of periods choosing price $\mathbf{x}$, i.e., $T_{\mathbf{x}} = \sum_{m \in \mathcal{M}_{\mathbf{x}}} I_m L\tau$. Then,

$$\sum_{m=1}^{M} \mathbb{E}\left[ \left( U_m(\mathbf{x}_m) - L_m(\mathbf{x}_m) \right) I_m L\tau \right] = \sum_{\mathbf{x} \in \mathcal{X}} \sum_{m \in \mathcal{M}_{\mathbf{x}}} \mathbb{E}\left[ U_m(\mathbf{x}) - L_m(\mathbf{x}) \right] I_m L\tau \tag{20a}$$

$$= \sum_{\mathbf{x} \in \mathcal{X}} \sum_{m \in \mathcal{M}_{\mathbf{x}}} 2\mathbb{E}\left[ \mathbf{Rad}_m(\mathbf{x}) \right] I_m L\tau \tag{20b}$$

$$\leq \sum_{\mathbf{x} \in \mathcal{X}} O\left( J\sqrt{|\mathcal{M}_{\mathbf{x}}| T_{\mathbf{x}} \log(J^{1/4}T)} \right) \tag{20c}$$

$$= O\left( J\sqrt{\log(J^{1/4}T)} \right) \sum_{\mathbf{x} \in \mathcal{X}} \sqrt{|\mathcal{M}_{\mathbf{x}}| T_{\mathbf{x}}} \tag{20d}$$

$$\leq O\left( J\sqrt{\log(J^{1/4}T) \log T} \right) \sum_{\mathbf{x} \in \mathcal{X}} \sqrt{T_{\mathbf{x}}} \tag{20e}$$

$$\leq O\left( J\sqrt{XT \log T \log(J^{1/4}T)} \right), \tag{20f}$$

where (20c) by Lemma 2, (20e) is derived by the number of the batches, and (20f) follows Jensen's inequality. The number of batches is $\log_2\left( 1 + \frac{T - 8X \log(J^{1/4}T)}{2L \log(T)^2} \right) = O\left( \log(T) \right)$.

Below we analyze the first term of (19), and the same logic applies to the third term, of which we omit the details here.

$$\sum_{m=1}^{M} \mathbb{E}\Big[\big(r(\vec{\alpha}_{\tilde{\mathbf{x}}}, \tilde{\mathbf{x}}) - U_m(\tilde{\mathbf{x}})\big)\Big] I_m L\tau \tag{21a}$$

$$\leq \sum_{m=1}^{M} \mathbb{E}\Big[\big(r(\vec{\alpha}_{\tilde{\mathbf{x}}}, \tilde{\mathbf{x}}) - U_m(\tilde{\mathbf{x}})\big)\mathbb{1}\big(r(\vec{\alpha}_{\tilde{\mathbf{x}}}, \tilde{\mathbf{x}}) > U_m(\tilde{\mathbf{x}})\big)\Big] I_m L\tau \tag{21b}$$

$$\leq \sum_{m=1}^{M} \mathbb{E}\Big[r(\vec{\alpha}_{\tilde{\mathbf{x}}}, \tilde{\mathbf{x}})\mathbb{1}\big(r(\vec{\alpha}_{\tilde{\mathbf{x}}}, \tilde{\mathbf{x}}) > U_m(\tilde{\mathbf{x}})\big) - U_m(\tilde{\mathbf{x}})\mathbb{1}\big(r(\vec{\alpha}_{\tilde{\mathbf{x}}}, \tilde{\mathbf{x}}) > U_m(\tilde{\mathbf{x}})\big)\Big] I_m L\tau \tag{21c}$$

$$= \sum_{m=1}^{M} r(\vec{\alpha}_{\tilde{\mathbf{x}}}, \tilde{\mathbf{x}})\mathbb{P}\big(r(\vec{\alpha}_{\tilde{\mathbf{x}}}, \tilde{\mathbf{x}}) > U_m(\tilde{\mathbf{x}})\big) I_m L\tau - \sum_{m=1}^{M} \mathbb{E}\Big[U_m(\tilde{\mathbf{x}})\mathbb{1}\big(r(\vec{\alpha}_{\tilde{\mathbf{x}}}, \tilde{\mathbf{x}}) > U_m(\tilde{\mathbf{x}})\big)\Big] I_m L\tau \tag{21d}$$

$$\leq \sum_{m=1}^{M} r(\vec{\alpha}_{\tilde{\mathbf{x}}}, \tilde{\mathbf{x}})\mathbb{P}\big(r(\vec{\alpha}_{\tilde{\mathbf{x}}}, \tilde{\mathbf{x}}) > U_m(\tilde{\mathbf{x}})\big) \cdot I_m L\tau \tag{21e}$$

$$\leq O\left(\frac{1}{T^3}\right), \tag{21f}$$

where (21e) is because $U_m(\tilde{\mathbf{x}}) > 0$ by Definition 2, and (21f) follows Proposition 2.

Given that $O\big(n_{Mj}(\mathbf{x})\big) = O(T_{\mathbf{x}})$, for any product $j \in \mathcal{J}_{\text{on}}(\mathbf{x})$, we have $\sum_{m=1}^{M} O(L\tau) = O\big(L(\log(T))^3\big)$. Combining the three terms together,

$$\overline{\text{Regret}}_{\text{Learning}}(\text{RNRM-UCB}, T) = \sum_{m=1}^{M} \mathbb{E}\Big[\Delta(\mathbf{x}_m) I_m L\tau\Big] + \sum_{m=1}^{M} O(L\tau)$$

$$= O\left(J\sqrt{XT \log T \log(J^{1/4}T)}\right) + O\left(\frac{1}{T^3}\right) + O\big(L(\log(T))^3\big) \tag{22}$$

$$= O\left(J\sqrt{XT \log T \log(J^{1/4}T)}\right).$$

**Total Regret of RNRM-UCB algorithm.** By adding up the relaxed regret during the Warm-up and Learning Phases, we have the regret of RNRM-UCB algorithm as (typically we always have $T \gg X$):

$$\overline{\text{Regret}}(\text{RNRM-UCB}, T) = \overline{\text{Regret}}_{\text{Warm-up}}(\text{RNRM-UCB}, T) + \overline{\text{Regret}}_{\text{Learning}}(\text{RNRM-UCB}, T) \tag{23a}$$

$$= O(X \log(J^{1/4}T)) + O\left(J\sqrt{XT \log T \log(J^{1/4}T)}\right) \tag{23b}$$

$$= \tilde{O}(J\sqrt{XT}). \tag{23c}$$

**Q.E.D.**

*Proof of Theorem 2.*

The analysis here is similar to that in the proof of Theorem 1. We first bound the Bayesian regret of the Warm-up Phase and then for the Learning Phase.

**Bayesian regret in the Warm-up Phase.** Similarly, we can bound the Bayesian regret in the Warm-up Phase as

$$\overline{\text{BR}}_{\text{Warm-up}}(\text{RNRM-TS}, T) = O(X \log(J^{1/4}T)). \tag{24}$$

**Bayesian regret in the Learning Phase.** In batch $m$, the chosen price $\mathbf{x}_m$ and the state-independent optimal price $\tilde{\mathbf{x}}$ are identically distributed conditional on $\mathcal{F}_m$, which indicates that $\mathbb{E}\big[U_m(\mathbf{x}_m) - U_m(\tilde{\mathbf{x}})|\mathcal{F}_m\big] = 0$ (e.g., see the proof of Lemma 3.7 in Slivkins (2019)). Thus, the Bayesian regret in

batch $m$ is:

$$\overline{\text{BR}}_m \tag{25a}$$

$$= \underset{\{\alpha_j(\mathbf{x})\}_{j \in \mathcal{J}} \sim \text{priori}}{\mathbb{E}} \left[ \mathbb{E}\big[ r(\vec{\alpha}_{\tilde{\mathbf{x}}}, \tilde{\mathbf{x}}) I_m L\tau - Y_m^{\text{RNRM-TS}}(\mathbf{x}_m) | \mathcal{F}_m \big] \right] \tag{25b}$$

$$= \underset{\{\alpha_j(\mathbf{x})\}_{j \in \mathcal{J}} \sim \text{priori}}{\mathbb{E}} \left[ \mathbb{E}\big[ I_m L\tau U_m(\mathbf{x}_m) - Y_m^{\text{RNRM-TS}}(\mathbf{x}_m) + r(\vec{\alpha}_{\tilde{\mathbf{x}}}, \tilde{\mathbf{x}}) I_m L\tau - I_m L\tau U_m(\tilde{\mathbf{x}}) | \mathcal{F}_m \big] \right] \tag{25c}$$

$$= \underset{\{\alpha_j(\mathbf{x})\}_{j \in \mathcal{J}} \sim \text{priori}}{\mathbb{E}} \left[ \mathbb{E}\big[ I_m L\tau U_m(\mathbf{x}_m) - Y_m^{\text{RNRM-TS}}(\mathbf{x}_m) | \mathcal{F}_m \big] \right] \tag{25d}$$

$$+ \underset{\{\alpha_j(\mathbf{x})\}_{j \in \mathcal{J}} \sim \text{priori}}{\mathbb{E}} \left[ \mathbb{E}\big[ r(\vec{\alpha}_{\tilde{\mathbf{x}}}, \tilde{\mathbf{x}}) I_m L\tau - I_m L\tau U_m(\tilde{\mathbf{x}}) | \mathcal{F}_m \big] \right] \tag{25e}$$

$$= \mathbb{E}\big[ I_m L\tau U_m(\mathbf{x}_m) - Y_m^{\text{RNRM-TS}}(\mathbf{x}_m) \big] + \mathbb{E}\big[ r(\vec{\alpha}_{\tilde{\mathbf{x}}}, \tilde{\mathbf{x}}) I_m L\tau - I_m L\tau U_m(\tilde{\mathbf{x}}) \big] \tag{25f}$$

$$\leq \mathbb{E}\big[ I_m L\tau U_m(\mathbf{x}_m) - I_m L\tau \cdot r(\vec{\alpha}_{\mathbf{x}_m}, \mathbf{x}_m) + O(L\tau) \big] + \mathbb{E}\big[ I_m L\tau \cdot r(\vec{\alpha}_{\tilde{\mathbf{x}}}, \tilde{\mathbf{x}}) - I_m L\tau U_m(\tilde{\mathbf{x}}) \big] \tag{25g}$$

$$= I_m L\tau \Big( \mathbb{E}\big[ U_m(\mathbf{x}_m) - r(\vec{\alpha}_{\mathbf{x}_m}, \mathbf{x}_m) \big] + \mathbb{E}\big[ r(\vec{\alpha}_{\tilde{\mathbf{x}}}, \tilde{\mathbf{x}}) - U_m(\tilde{\mathbf{x}}) \big] \Big) + O(L\tau), \tag{25h}$$

where (25g) is derived following the analysis of nonstationarity in (16).

Derived from Proposition 2, for some $\gamma > 0$, two properties hold (see, e.g., Slivkins 2019):

$$\mathbb{E}[(r(\vec{\alpha}_{\mathbf{x}}, \mathbf{x}) - U_m(\mathbf{x}))^+] \leq \frac{\gamma}{XT} \qquad \text{and} \qquad \mathbb{E}[(L_m(\mathbf{x}) - r(\vec{\alpha}_{\mathbf{x}}, \mathbf{x}))^+] \leq \frac{\gamma}{XT}. \tag{26}$$

Following these two properties of the stationary revenue rate in (26), we have

$$\mathbb{E}\big[ r(\vec{\alpha}_{\tilde{\mathbf{x}}}, \tilde{\mathbf{x}}) - U_m(\tilde{\mathbf{x}}) \big] \leq \mathbb{E}\big[ (r(\vec{\alpha}_{\tilde{\mathbf{x}}}, \tilde{\mathbf{x}}) - U_m(\tilde{\mathbf{x}}))^+ \big] \leq \sum_{\mathbf{x} \in \mathcal{X}} \mathbb{E}\big[ (r(\vec{\alpha}_{\mathbf{x}}, \mathbf{x}) - U_m(\mathbf{x}))^+ \big] \leq \frac{\gamma}{T}, \tag{27}$$

and similarly,

$$\begin{aligned}
\mathbb{E}\big[ U_m(\mathbf{x}_m) - r(\lambda_{\mathbf{x}_m}, \mu_{\mathbf{x}_m}, \mathbf{x}_m) \big] &= \mathbb{E}\big[ 2 \cdot \mathbf{Rad}_m(\mathbf{x}_m) + L_m(\mathbf{x}_m) - r(\vec{\alpha}_{\mathbf{x}_m}, \mathbf{x}_m) \big] \\
&\leq \mathbb{E}\big[ 2 \cdot \mathbf{Rad}_m(\mathbf{x}_m) \big] + \mathbb{E}\big[ (L_m(\mathbf{x}) - r(\vec{\alpha}_{\mathbf{x}_m}, \mathbf{x}_m))^+ \big] \\
&\leq \mathbb{E}\big[ 2 \cdot \mathbf{Rad}_m(\mathbf{x}_m) \big] + \sum_{\mathbf{x} \in \mathcal{X}} \mathbb{E}\big[ (L_m(\mathbf{x}) - r(\vec{\alpha}_{\mathbf{x}_m}, \mathbf{x}_m))^+ \big] \\
&\leq \mathbb{E}\big[ 2 \cdot \mathbf{Rad}_m(\mathbf{x}_m) \big] + \frac{\gamma}{T}.
\end{aligned} \tag{28}$$

Summing over all batches $m, \forall m = 1, \ldots, M$, we have the cumulative relaxed Bayesian regret by the end of the Learning Phase as:

$$\overline{BR}_{\text{Learning}}(\pi_{\text{RNRM-TS}}, T) \tag{29a}$$

$$= \sum_{m=1}^{M} \overline{BR}_m \tag{29b}$$

$$= \sum_{m=1}^{M} \left( I_m L \tau \left( \mathbb{E}\left[ U_m(\mathbf{x}_m) - r(\vec{\alpha}_{\mathbf{x}_m}, \mathbf{x}_m) \right] + \mathbb{E}\left[ r(\vec{\alpha}_{\tilde{\mathbf{x}}}, \tilde{\mathbf{x}}) - U_m(\tilde{\mathbf{x}}) \right] \right) + O(L\tau) \right) \tag{29c}$$

$$\leq \sum_{m=1}^{M} \left( I_m L \tau \left( 2\frac{\gamma}{T} + 2\mathbb{E}\left[ \mathbf{Rad}_m(\mathbf{x}_m) \right] \right) \right) + O\left( \tau \log(T) \right) \tag{29d}$$

$$\leq 2 \sum_{m=1}^{M} I_m L \tau \frac{\gamma}{T} + 2 \sum_{\mathbf{x} \in \mathcal{X}} \sum_{m \in \mathcal{M}_{\mathbf{x}}} I_m L \tau \mathbb{E}\left[ \mathbf{Rad}_m(\mathbf{x}) \right] + O\left( \tau \log(T) \right) \tag{29e}$$

$$= 2\gamma + \leq \sum_{\mathbf{x} \in \mathcal{X}} O\left( J\sqrt{|\mathcal{M}_{\mathbf{x}}|T_{\mathbf{x}} \log(J^{1/4}T)} \right) + O\left( \tau \log(T) \right) \tag{29f}$$

$$\leq \sum_{\mathbf{x} \in \mathcal{X}} O\left( J\sqrt{|\mathcal{M}_{\mathbf{x}}|T_{\mathbf{x}} \log(J^{1/4}T)} \right) + O\left( (\log(T))^3 \right) \tag{29g}$$

$$= \sum_{\mathbf{x} \in \mathcal{X}} O\left( J\sqrt{|\mathcal{M}_{\mathbf{x}}|T_{\mathbf{x}} \log(J^{1/4}T)} \right) \tag{29h}$$

$$= O\left( J\sqrt{XT \log T \log(J^{1/4}T)} \right), \tag{29i}$$

where (29f) is by Lemma 2 and (29g) is by Jensen's inequality.

**Regret of RNRM-TS algorithm.** By adding up the relaxed regret during the Warm-up and Learning Phases, we have the regret of RNRM-TS algorithm as:

$$\overline{BR}(\text{RNRM-TS}, T) = \overline{BR}_{\text{Warm-up}}(\text{RNRM-TS}, T) + \overline{BR}_{\text{Learning}}(\text{RNRM-TS}, T) \tag{30a}$$

$$= O(X \log(J^{1/4}T)) + O\left( J\sqrt{XT \log T \log(J^{1/4}T)} \right) \tag{30b}$$

$$= \tilde{O}(J\sqrt{XT}). \tag{30c}$$

**Q.E.D.**

**Lemma 2.** *For any price* $\mathbf{x} \in \mathcal{X}$,

$$\sum_{m \in \mathcal{M}_{\mathbf{x}}} I_m L \tau \mathbb{E}\left[ \mathbf{Rad}_m(p) \right] = O\left( J\sqrt{|\mathcal{M}_{\mathbf{x}}|T_{\mathbf{x}} \log(J^{1/4}T)} \right). \tag{31}$$

*Proof of Lemma 2.* Plugging the definition of $\mathbf{Rad}_m(\mathbf{x})$, $\forall m = 1, \dots, M$, one has:

$$\sum_{m \in \mathcal{M}_\mathbf{x}} I_m L\tau \mathbb{E}\big[\mathbf{Rad}_m(\mathbf{x})\big] = O\left(\sqrt{\log(J^{1/4}T)}\right) \sum_{m \in \mathcal{M}_\mathbf{x}} I_m L\tau \cdot O\left(\sum_{j \in \mathcal{J}_{\text{on}}(\mathbf{x})} \frac{1}{\sqrt{n_{mj}(\mathbf{x})}}\right) \tag{32a}$$

$$\leq O\left(\sqrt{\log(J^{1/4}T)}\right) \sum_{m \in \mathcal{M}_\mathbf{x}} I_m L\tau \cdot O\left(\sum_{j \in \mathcal{J}_{\text{on}}(\mathbf{x})} \frac{1}{\sqrt{I_m L\tau}}\right) \tag{32b}$$

$$= O\left(\sqrt{\log(J^{1/4}T)}\right) \sum_{m \in \mathcal{M}_\mathbf{x}} I_m L\tau \cdot O\left(\frac{J}{\sqrt{I_m L\tau}}\right) \tag{32c}$$

$$= O\left(J\sqrt{\log(J^{1/4}T)}\right) \sum_{m \in \mathcal{M}_\mathbf{x}} \sqrt{I_m L\tau} \tag{32d}$$

$$\leq O\left(J\sqrt{\log(J^{1/4}T)}\right) \cdot \sqrt{|\mathcal{M}_\mathbf{x}| \sum_{m \in \mathcal{M}_\mathbf{x}} I_m L\tau} \tag{32e}$$

$$= O\left(J\sqrt{|\mathcal{M}_\mathbf{x}|T_\mathbf{x} \log(J^{1/4}T)}\right) \tag{32f}$$

$$\tag{32g}$$

where and (32b) is by $n_{mj}(\mathbf{x}) = O(I_m L\tau)$, $\forall j \in \mathcal{J}_{\text{on}}(\mathbf{x})$ and (32e) is by Jensen's inequality.

**Lemma 3.** *(Coupling probability of M/M/c queue starting from empty state, Proposition 4 in Jia et al. 2020) For $\tau = (\log T)^2$, if an M/M/c queue starts from empty state at time $0$, then the probability that the system reaches the steady-state within $\tau$ is bounded by $1 - \frac{2}{T^2}$.*

**Lemma 4.** *(Coupling probability of M/M/c queue when price changes, Proposition 5 in Jia et al. 2020) For $\tau = (\log T)^2$, if an M/M/c queue starts from steady-state and changes system parameter at time $0$, then the probability that the system reaches the steady-state within $2\tau$ is bounded by $1 - \frac{4}{T^2}$.*

# F   Service Duration Dependent Payment Model

We consider a new variant payment model where the customer pays the product's per-unit-time price multiplied by its total service duration. The firm then needs to decide the per-unit-time price for each product $j \in \mathcal{J}$. Under this payment setting, we modify the definition of the steady-state revenue rate, derive a new concentration bound of historical observations and actual system parameters, and then correspondingly develop a new UCB for the revenue rate of each price. With those modified definitions, we present a new RNRM-UCB variant and show that it achieves the same regret bound.

**Data collection.** Similar to $n_j$, $\hat{d}_{jk}(\mathbf{x})$, and $\bar{d}_j(\mathbf{x})$, we also collect service time observations over resources. We use $n_i^r(\mathbf{x})$ to denote the number of customers who successfully consume resource $i$ under price $\mathbf{x}$. We use $\hat{g}_{ik}(\mathbf{x})$ to denote the service time of customer $k$ over resource $i$ under the price $\mathbf{x}$. We use $\bar{g}_i(\mathbf{x})$ to denote the empirical mean of the service time and $\bar{g}_i(\mathbf{x}) = \sum_{k=1}^{n_i^r(\mathbf{x})} \hat{g}_{ik}(\mathbf{x})/n_i^r(\mathbf{x})$, for $i \in \mathcal{I}$. We further assume that there is a known upper bound $r_{\max} = \max_{x,i,j} \alpha_j/\mu_i$, $n_j^{\min}(\mathbf{x}) = \min\{n_j(\mathbf{x}), \min_{i:a_{ji}=1} n_i^s(\mathbf{x})\}$.

**Revenue rate.** Under this duration-dependent payment model, we offer unit price $\mathbf{x} \in \mathbb{R}^J$ to products $\mathcal{J}$. Customer $k$ who requests to use product $j$ pays the per-unit-time price multiplied by the total service duration, i.e., $x_j \sum_{i:a_{ji}=1} \hat{g}_{ik}(\mathbf{x})$, to the firm. Different than the lump sum payment model, the steady-state revenue rate for this setting is $r^v(\mathbf{x}) = \sum_{j \in \mathcal{J}_{\text{on}}(\mathbf{x})} \alpha_j x_j \sum_{i:a_{ji}=1} 1/\mu_i(\mathbf{x})$.

**Proposition 5.** *For any price $\mathbf{x} \in \mathcal{X}$, if the number of arrival time observations for each turn-on product $n_j(\mathbf{x}) \geq 8\log(L^{\frac{1}{4}}J^{\frac{1}{4}}T)$ for $j \in \mathcal{J}_{on}(\mathbf{x})$ and the number of service time observations for each resource $n_i^s(\mathbf{x}) \geq 8\log(L^{\frac{1}{4}}J^{\frac{1}{4}}T)$ $\forall i \in \{i \in \mathcal{I} : \sum_{j \in \mathcal{J}_{on}(\mathbf{x})} a_{ji} \geq 1\}$, then we have*

$$\mathbb{P}\left(\left| r^v(\mathbf{x}) - \sum_{j \in \mathcal{J}_{on}(\mathbf{x})} \frac{x_j \sum_{i:a_{ji}=1} \bar{g}_i(\mathbf{x})}{\bar{d}_j(\mathbf{x})} \right| \leq \sum_{j \in \mathcal{J}_{on}(\mathbf{x})} \frac{\sum_{i:a_{ji}=1} (\frac{\bar{g}_i(\mathbf{x})}{\bar{d}_j(\mathbf{x})} + r_{max})x_j\sqrt{32\log(L^{\frac{1}{4}}J^{\frac{1}{4}}T)}}{\sqrt{n_j^{min}(\mathbf{x})}} \right) \geq 1 - \frac{4}{T^4},$$

*where $r_{max} = \max_{\mathbf{x},i,j} \alpha_j(\mathbf{x})/\mu_i(\mathbf{x})$, and $n_j^{min}(\mathbf{x}) = \min\{n_j(\mathbf{x}), \min_{i:a_{ji}=1} n_i^s(\mathbf{x})\}$, and $L$ is the number of layers.*

*Proof of Proposition 5.*

The above probability can be bounded by the product of the following probabilities from below. Firstly we can bound that for $i, j$ with $a_{ji} = 1$:

$$\mathbb{P}\left(\left|\frac{\alpha_j}{\mu_i(\mathbf{x})} - \frac{\bar{g}_i(\mathbf{x})}{\bar{d}_j(\mathbf{x})}\right| \leq \frac{(\frac{\bar{g}_i(\mathbf{x})}{\bar{d}_j(\mathbf{x})} + r_{max})\sqrt{32\log(L^{\frac{1}{4}}J^{\frac{1}{4}}T)}}{\sqrt{\min\{n_j(\mathbf{x}), n_i^r(\mathbf{x})\}}}\right)$$

$$\geq \mathbb{P}\left(\left|\frac{\alpha_j}{\mu_i(\mathbf{x})} - \alpha_j\bar{g}_i(\mathbf{x}) + \alpha_j\bar{g}_i(\mathbf{x}) - \frac{\bar{g}_i(\mathbf{x})}{\bar{d}_j(\mathbf{x})}\right| \leq \frac{(\frac{\bar{g}_i(\mathbf{x})}{\bar{d}_j(\mathbf{x})} + r_{max})\sqrt{32\log(L^{\frac{1}{4}}J^{\frac{1}{4}}T)}}{\sqrt{\min\{n_j(\mathbf{x}), n_i^r(\mathbf{x})\}}}\right)$$

$$\geq \mathbb{P}\left(\alpha_j\left|\frac{1}{\mu_i(\mathbf{x})} - \bar{g}_i(\mathbf{x})\right| + \bar{g}_i(\mathbf{x})\left|\alpha_j - \frac{1}{\bar{d}_j(\mathbf{x})}\right| \leq \frac{(\frac{\bar{g}_i(\mathbf{x})}{\bar{d}_j(\mathbf{x})} + r_{max})\sqrt{32\log(L^{\frac{1}{4}}J^{\frac{1}{4}}T)}}{\sqrt{\min\{n_j(\mathbf{x}), n_i^r(\mathbf{x})\}}}\right)$$

$$\geq \mathbb{P}\left(\alpha_j\left|\frac{1}{\mu_i(\mathbf{x})} - \bar{g}_i(\mathbf{x})\right| \leq \frac{r_{max}\sqrt{32\log(L^{\frac{1}{4}}J^{\frac{1}{4}}T)}}{\sqrt{n_i^r(\mathbf{x})}}\right) \cdot \mathbb{P}\left(\bar{g}_i(\mathbf{x})\left|\alpha_j - \frac{1}{\bar{d}_j(\mathbf{x})}\right| \leq \frac{\frac{\bar{g}_i(\mathbf{x})}{\bar{d}_j(\mathbf{x})}\sqrt{32\log(L^{\frac{1}{4}}J^{\frac{1}{4}}T)}}{\sqrt{n_j(\mathbf{x})}}\right)$$

$$\geq \mathbb{P}\left(\left|\frac{1}{\mu_i(\mathbf{x})} - \bar{g}_i(\mathbf{x})\right| \leq \frac{\sqrt{32\log(L^{\frac{1}{4}}J^{\frac{1}{4}}T)}}{\mu_i(\mathbf{x})\sqrt{n_i^r(\mathbf{x})}}\right) \cdot \mathbb{P}\left(\left|\alpha_j - \frac{1}{\bar{d}_j(\mathbf{x})}\right| \leq \frac{\sqrt{32\log(L^{\frac{1}{4}}J^{\frac{1}{4}}T)}}{\bar{d}_j(\mathbf{x})\sqrt{n_j(\mathbf{x})}}\right).$$

The first and second probabilities in the above equation can be bounded with the same technique as that we used in (7) with Lemma 1 and thus we have

$$\mathbb{P}\left(\left|\alpha_j - \frac{1}{\bar{d}_j(\mathbf{x})}\right| \leq \frac{\sqrt{32\log(L^{\frac{1}{4}}J^{\frac{1}{4}}T)}}{\bar{d}_j(\mathbf{x})\sqrt{n_j(\mathbf{x})}}\right) \geq 1 - \frac{2}{LJT^4}$$

and

$$\mathbb{P}\left(\left|\frac{1}{\mu_i(\mathbf{x})} - \bar{g}_i(\mathbf{x})\right| \leq \frac{\sqrt{32\log(L^{\frac{1}{4}}J^{\frac{1}{4}}T)}}{\mu_i(\mathbf{x})\sqrt{n_i^r(\mathbf{x})}}\right) \geq 1 - \frac{2}{LJT^4}.$$

Therefore, we have the above probability can be bounded by

$$\left(1 - \frac{2}{LJT^4}\right)^2 \geq 1 - \frac{4}{LJT^4}.$$

Then, for product $j \in \mathcal{J}_{on}(\mathbf{x})$, we can further bound

$$\mathbb{P}\left(\left|\sum_{i:a_{ji}=1}\frac{\alpha_j}{\mu_i(\mathbf{x})} - \sum_{i:a_{ji}=1}\frac{\bar{g}_i(\mathbf{x})}{\bar{d}_j(\mathbf{x})}\right| \leq \sum_{i:a_{ji}=1}\frac{(\frac{\bar{g}_i(\mathbf{x})}{\bar{d}_j(\mathbf{x})} + r_{max})\sqrt{32\log(L^{\frac{1}{4}}J^{\frac{1}{4}}T)}}{\sqrt{n_j^{min}(\mathbf{x})}}\right)$$

$$\geq \prod_{i:a_{ji}=1}\mathbb{P}\left(\left|\frac{\alpha_j}{\mu_i(\mathbf{x})} - \frac{\bar{g}_i(\mathbf{x})}{\bar{d}_j(\mathbf{x})}\right| \leq \frac{(\frac{\bar{g}_i(\mathbf{x})}{\bar{d}_j(\mathbf{x})} + r_{max})\sqrt{32\log(L^{\frac{1}{4}}J^{\frac{1}{4}}T)}}{\sqrt{n_j^{min}(\mathbf{x})}}\right)$$

$$\geq \left(1 - \frac{4}{LJT^4}\right)^{|\{i:a_{ji}=1\}|} \geq \left(1 - \frac{4}{LJT^4}\right)^L \geq 1 - \frac{4}{JT^4},$$

because $\sum_{i\in\mathcal{I}} a_{ji} \leq L \; \forall j \in \mathcal{J}$.

Now we are ready to derive the concentration bound.

$$
\mathbb{P}\left( \left| r^v(\mathbf{x}) - \sum_{j \in \mathcal{J}_{\mathrm{on}}(\mathbf{x})} \frac{x_j \sum_{i:a_{ji}=1} \bar{g}_i(\mathbf{x})}{\bar{d}_j(\mathbf{x})} \right| \leq \sum_{j \in \mathcal{J}_{\mathrm{on}}(\mathbf{x})} \frac{\sum_{i:a_{ji}=1} (\frac{\bar{g}_i(\mathbf{x})}{\bar{d}_j(\mathbf{x})} + r_{\max}) x_j \sqrt{32 \log(L^{\frac{1}{4}} J^{\frac{1}{4}} T)}}{\sqrt{n_j^{\min}(\mathbf{x})}} \right)
$$

$$
\geq \prod_{j \in \mathcal{J}_{\mathrm{on}}(\mathbf{x})} \mathbb{P}\left( \left| \sum_{i:a_{ji}=1} \frac{\alpha_j}{\mu_i(\mathbf{x})} - \sum_{i:a_{ji}=1} \frac{\bar{g}_i(\mathbf{x})}{\bar{d}_j(\mathbf{x})} \right| \leq \sum_{i:a_{ji}=1} \frac{(\frac{\bar{g}_i(\mathbf{x})}{\bar{d}_j(\mathbf{x})} + r_{\max}) \sqrt{32 \log(L^{\frac{1}{4}} J^{\frac{1}{4}} T)}}{\sqrt{n_j^{\min}(\mathbf{x})}} \right)
$$

$$
\geq \left( 1 - \frac{4}{J \cdot T^4} \right)^{|\mathcal{J}_{\mathrm{on}}(\mathbf{x})|} \geq \left( 1 - \frac{4}{J \cdot T^4} \right)^{J} \geq 1 - \frac{4J}{J \cdot T^4} = 1 - \frac{4}{T^4}.
$$

**Q.E.D.**

According to the above proposition, we can define the confidence radius as follows. (We use the superscript $v$ to denote this duration-dependent payment model.)

$$
\mathbf{Rad}_t^v(\mathbf{x}) = \sum_{j \in \mathcal{J}_{\mathrm{on}}(\mathbf{x})} \frac{\sum_{i:a_{ji}=1} (\frac{\bar{g}_i(\mathbf{x})}{\bar{d}_j(\mathbf{x})} + r_{\max}) x_j \sqrt{32 \log(L^{\frac{1}{4}} J^{\frac{1}{4}} T)}}{\sqrt{n_j^{\min}(\mathbf{x})}}.
$$

Therefore, the upper and lower confidence bound of the revenue rate associated with each price is

$$
U_t^v(\mathbf{x}) = \sum_{j \in \mathcal{J}_{\mathrm{on}}(\mathbf{x})} \frac{x_j \sum_{i:a_{ji}=1} \bar{g}_i(\mathbf{x})}{\bar{d}_j(\mathbf{x})} + \mathbf{Rad}_t^v(\mathbf{x}), \qquad L_t^v(\mathbf{x}) = \sum_{j \in \mathcal{J}_{\mathrm{on}}(\mathbf{x})} \frac{x_j \sum_{i:a_{ji}=1} \bar{g}_i(\mathbf{x})}{\bar{d}_j(\mathbf{x})} - \mathbf{Rad}_t^v(\mathbf{x}).
$$

**Algorithm design.** The algorithmic framework remains unchanged except for two modifications. (i) In Step 6, the algorithm also requires the number of service time observations to be at least $8 \log(L^{\frac{1}{4}} J^{\frac{1}{4}} T)$. (ii) In Steps 13 and 16, the algorithm uses the new UCB definition.

---

**Algorithm 3** RNRM-UCB (Duration-Dependent Payment Model.)

---

1: Input: $T$, $\mathcal{X}$, service network, turn-off prices.
2: Initialize: $L$, $\tau$, $I_m$, $M$, $m = 0$.
3: Warm-up Phase:
4: **for** $\mathbf{x} \in \mathcal{X}$ **do**
5:     Offer $\mathbf{x}$, record inter-arrival time $\hat{d}_{jk}(\mathbf{x})$ for arriving customers of product $j \in \mathcal{J}_{\mathrm{on}}(\mathbf{x})$.
6:     **if** $n_{mj}^{\min}(\mathbf{x}) \geq 8 \log(L^{\frac{1}{4}} J^{\frac{1}{4}} T)$ for $j \in \mathcal{J}_{\mathrm{on}}(\mathbf{x})$ **then**
7:         Continue.
8:     **end if**
9: **end for**
10: Compute and update UCB for all prices as defined in (2).
11: Learning Phase:
12: **for** $m = 1, \ldots, M$ **do**
13:     Choose $\mathbf{x}_m = \mathrm{argmax}_{\mathbf{x} \in \mathcal{X}} U_{m-1}^v(\mathbf{x})$.
14:     Offer $\mathbf{x}_m$ in batch $m$, i.e., for $I_m L \tau$ periods.
15:     Record inter-arrival time $\hat{d}_{jk}(\mathbf{x}_m)$ for arriving customers of product $j \in \mathcal{J}_{\mathrm{on}}(\mathbf{x}_m)$.
16:     Compute and update $U_m^v(\mathbf{x_m})$ .
17: **end for**

---

**Regret analysis.** We change the constraints (1d) in the LP benchmark to

$$
Y_{\mathbf{y}t} \leq \sum_{\mathbf{x} \in \mathcal{X}} \pi_{\mathbf{y}t}^{\mathbf{x}} \cdot \sum_{j \in \mathcal{J}_{\mathrm{on}}(\mathbf{x})} \alpha_j x_j \sum_{i:a_{ji}=1} \frac{1}{\mu_i(\mathbf{x})} \quad \forall \mathbf{y} \in \{0, 1, \ldots, \infty\}^I, \ \forall t = 1, \ldots, T.
$$

Under this duration-dependent payment setting, we can follow the same analysis technique to show that this LP model also gives a valid upper bound of the optimal revenue. The results of Propositions 1, 3, and 4 and Theorem 1 remain the same for this duration-dependent payment model.