# OpenReview forum: "Online Learning and Pricing for Network Revenue Management with Reusable Resources"
_NeurIPS.cc/2022/Conference — NeurIPS 2022 Accept_

### Official Review · Reviewer_dRYY · 2022-07-03

**Rating:** 7
**Confidence:** 4
**Soundness:** 4 excellent
**Presentation:** 4 excellent
**Contribution:** 2 fair

**Summary:**

The paper addresses the general problem of setting product prices in a multi-resource and multi-product service environment so that these prices result in nearly optimal total revenue that is generated, assuming rapid adjustment of demand in response to service price changes.  In this generality, and as stated in the abstract and in the introductory sections, the model is applicable to a host of different settings ranging from computing clouds, manufacturing, and several other service platforms, both virtual and physical, also listed in the introduction.  A solution is provided via a bandit process which minimizes regret, that is the gap between what is achieved and the optimal prescient solution.  The methodology crucially depends on fitting the model to Jackson networks and along the way not only sophisticated results (such as finite-time mixing time of Jackson networks) are derived, proved and used but several difficult theorems also stated and proved.  A numerical example is provided at the end to demonstrate applicability of the results in practice.

**Questions:**


There is some difficulty in understanding the process of demand-price the author(s) have in mind.  Are prices computed for "resources" and then advertised for "services" which use these resources in a linear fashion? Also the following statements need clarification.

Line 68: In the first interval, the system completes serving the previous customers and reaches the steady-state under the new price.
69.
Q:  Only those customers?  Any new customer arriving in the next interval are queued or are they rejected and cleared from the system until new prices are advertised?

Line 69: In the second interval, the system maintains the steady-state (exploiting the new price).
Q: By "maintain" the author(s) mean the customers for the next batch wait long enough for the system go past the transient phase and enter equilibrium?  What happens during this time to newly arriving customers?  What prices are advertised during this time?

Line 70: This batched framework not only allows for system stabilization (under new prices) but also allows for infrequent O(log T) price changes, which is more implementable in practical settings.

Q: Why is price change not O(T) if T is selected to be the time during which prices don't change?

Line 133: Customers arrive following a price-dependent multivariate Poisson process with a total rate alpha(x), which is unknown to the decision maker a priori.

Q: Has demand-price being multivariate Poisson been observed in any real setting?  Otherwise is this is just for mathematical convenience?  This reviewer has not seen demand-price to be a Poisson-process before unless the mean demand value of the Poisson process is some (elastic/inelastic) function of resource price.  Btw, demand is going to be a function of *service* price and not resource price and the relationship between service price and resource price has not been discussed, as far as this reviewer can see.

Line 141: We use a J-dimensional vector.
Q: An I-dimensional vector?

Line 144: \pi (y; t) in \Xi for period t when there are y customers in the system at the beginning of period t, to maximize the expected cumulative revenue Y^\pi.
Q: Why is price only a function of state (y in R^+I) at the *beginning* of each period? Does demand for each period arrive instantaneously at the beginning when resource (should be service) prices are announced and no more demand arrives till the next time period?  Also, prices are stated to be for resources but customers ask for services so is the price of service/product simply sum of resource used?  This is of course true of costs but need not hold for prices. If so, this needs to be explicitly mentioned.


**Limitations:**


The biggest hurdle this reviewer sees is in the list of 5 assumptions.  Let's review these together.

Assumption 2. This is not really justifiable.  Many cloud or manufacturing tasks require specific order of processes which are task/service dependent and cannot be ordered the same way for all services.  There is huge loss of generality in making this assumption.

Assumption 3. Like assumption 2, this one is also extremely restrictive.  If we list each service and its resources, then we see a clear dependence between use of resources by services.  The use of the term "customer" is misleading here since customers ask for services and services have a specific set/sequence of resources which are interdependent.

It is very clear that assumptions 2 and 3 are leveraged to make the Jackson Network solution applicable to this problem and not because the problem even by some stretch of the imagination happens to satisfy the said assumptions.  If there are specific services (cloud/manufacturing) for which these assumptions are not restrictive, then these should be explicitly stated.

Assumption 4.  This assumption is unnatural but unlike assumptions 2 and 3, this one can be remedies by saying that resource prices are selected to be sufficiently high so that all resulting queues remain stable.  This is not unreasonable.  It just puts lower bounds on prices.
The statement "is stable under any price" should therefore be qualified.

Assumption 5.  This one even reads like a technical requirement not justified by any known services. Unless of course, the author(s) give specific examples from real cases.

**Strengths And Weaknesses:**


The problem setting considered in this submission is very broad, the quality of writing and the clarify of the discussion is also very high.  This paper would furnish a major advance in understanding multi-resource revenue management if the applicability of the results were as high as initially aimed for.  But despite derivation of advanced, original and complex theorems, the claims envisioned in the early part of this submission are unfortunately not realized.  This is primarily because a series of progressively untenable assumptions are made in Section 2 and these are so restrictive and so targeted to make the model fit a Jacksonian framework that it is highly doubtful if the bandit-based heuristic solution is useful for *any* of the many settings the author(s) described in the introduction.  This reviewer perfectly understand the nearly unsurmountable difficulties which would be encountered without making some simplifying assumptions in such a context.  But on the other hand, the restrictiveness of the underlying assumptions are too broad for the model to be tenable.  These issues are discussed in detail below.  The only way out of this dilemma is for the author(s) to show that the methodology works reasonably well or better than existing solutions *despite* the restrictiveness of the assumptions, ie demonstrate some kind of insensitivity to the restrictive assumptions made for analytical tractability.  If there is evidence for this, then this paper represents a significant contribution to the literature on network revenue management.  But Section 6 is too terse and too brief to convince this reviewer of this possibility.

---

> ### Author Response · Authors · 2022-08-01
> **Response to Reviewer dRYY - Part 1**
>
> We sincerely thank the reviewer for reading our manuscript carefully and giving numerous insightful comments. We have uploaded a revised manuscript to address your comments with highlighted colors.
>
> >**Comment 1.** Assumption 2. This is not really justifiable. Many cloud or manufacturing tasks require specific order of processes which are task/service dependent and cannot be ordered the same way for all services. There is huge loss of generality in making this assumption.
>
> **Response:** Thank you for question. We respectfully argue that Assumption 2 is a mild assumption that prevents cyclic graphs. This encompasses any tree-structured graphs and most acyclic graphs without bidirectional edges. To this end, we have provided a concrete example in the revised manuscript. Since it contains pictures and detailed descriptions, we would very much appreciate if you can quickly look at our example in the revised manuscript.
>
> >**Comment 2.** Assumption 3. Like assumption 2, this one is also extremely restrictive. If we list each service and its resources, then we see a clear dependence between use of resources by services. The use of the term ``customer" is misleading here since customers ask for services and services have a specific set/sequence of resources which are interdependent.
>
> **Response:** Thank you for question and sorry for causing some confusion about this assumption. This assumption simply allows us to write the product demand in the ``product-form'' of associated resource routing probabilities. In the revised manuscript, we drop this assumption and write our product demand processes more explicitly:
>
> **(Product-form demand.)** Any given price vector $\mathbf{x}$ induces a (Markovian) routing matrix $P(\mathbf{x})$ where its $(i,i')$th entry $p_{i,i'}$ represents the product demand portion who will consume resource $i'$ next out of those who consume resource $i$. Then the demand rate for any product $j\in \mathcal{J}$, represented by a sequence of resources ${s, i_1, i_2, \ldots, i_n, t}$, is given by
> $\alpha_j = \alpha(\mathbf{x}) p_{s,i_1}p_{i_1,i_2}\ldots p_{i_{n-1},i_n}p_{i_{n},t},$
> which means that the demand rate for product $j$ is equal to the total demand rate $\alpha(\mathbf{x})$ times a sequence of routing probabilities of $P(\mathbf{x})$, under posted price vector $\mathbf{x}$.
>
> We use Figure 2 in the Appendix to demonstrate a concrete example. With Assumptions 2 and 3, the service system under any posted price $\mathbf{x} \in \mathcal{X}$ can be reduced to a Jackson Network consisting with $I$ number of M/M/c queues (when the system only contains customers arriving under this price, i.e., after finishing serving other customers arriving under the previously posted prices).  The main idea is that there exists a mapping $\mathbf{x} \mapsto \alpha(\mathbf{x}), P(\mathbf{x})$ that gives rise to product demand pattern. Note that our assumptions are mild in the sense that this mapping can be nonparametric. The decision maker does not know this underlying mapping and needs to learn it while maximizing revenue on the fly.
>
> We hope that our revised assumption and example alleviate your concerns regarding the applicability of our model.
>
> >**Comment 3.** It is very clear that assumptions 2 and 3 are leveraged to make the Jackson Network solution applicable to this problem and not because the problem even by some stretch of the imagination happens to satisfy the said assumptions. If there are specific services (cloud/manufacturing) for which these assumptions are not restrictive, then these should be explicitly stated.
>
> **Response:** As you mentioned in your general assessment, analyzing any sequential learning problems with a network structure is extremely challenging but that is why there is almost a blank space in the literature. This product-form representation in the underlying network is realistic (see our revised example) and sufficiently tractable for us to carry out a formal and rigorous analysis. We hope that you agree that our results significantly contribute to the current understanding of network learning problems.
>
> >**Comment 4.** Assumption 4. This assumption is unnatural but unlike Assumptions 2 and 3, this one can be remedies by saying that resource prices are selected to be sufficiently high so that all resulting queues remain stable. This is not unreasonable. It just puts lower bounds on prices. The statement ``is stable under any price" should therefore be qualified.
>
> **Response:** Thank you for your suggestion. Indeed, putting lower bounds on prices justifies this assumption, and we have included it in the revised manuscript.

---

> > ### Comment · Reviewer_dRYY · 2022-08-03
> > **Part 1**
> >
> > I thank the authors for their thoughtful and detailed answers.
> >
> > Response to Comment 1:  OK, you're referring to a routing matrix and your worked out example helps.  Many cloud services do have A->B->C->B->Z routes or even self-loops (eg the bib task in Overleaf) and your model does not allow for this but I agree now Assumption 2 is not as restrictive as I stated.
> >
> > Response to Comment 2: Your re-write addressed this well. Thank you.
> >
> > Response to Comment 3: "... analyzing any sequential learning problems with a network structure is extremely challenging that is why there is almost a blank space in the literature".  This is true but I remain skeptical that your model filled any significant real gaps.  A model can help predict, optimize or reduce computations via closed-form solutions.  Which of these gaps would you say your work has now bridged?  What's your evidence for this response?
> >
> > Response to Comment 4:  Sure thing.

---

> > > ### Author Response · Authors · 2022-08-05
> > > **Response to Reviewer dRYY's Reply**
> > >
> > > We thank the reviewer for his/her prompt reply and favorable response. We are glad that you are now satisfied with our (re-stated) assumptions. We want to reiterate our contributions here. From the modeling perspective, we believe that we are the first to model this complex sequential network pricing problem via connecting to the celebrated Jackson Network. It allows for joint learning (of the rates and routing matrices) and optimization (of the total expected revenue) for a general class of network revenue management problems. From the algorithmic perspective, we propose a batched bandit algorithm with mixing time analysis on (price-dependent) Jackson Network. The algorithm is polynomial in running time and space, admits a near-optimal regret bound, and also demonstrates strong numerical performance. The high-probability coupling result of the Jackson Network is also of independent interest, since this finite-sample bound could be proven useful for designing learning algorithms for other complex stochastic networks. Again, thank you very much for your careful review and reply. Do not hesitate to ask us further questions if they occur.

---

> ### Author Response · Authors · 2022-08-01
> **Response to Reviewer dRYY - Part 2**
>
> >**Comment 5.** Assumption 5. This one even reads like a technical requirement not justified by any known services. Unless of course, the author(s) give specific examples from real cases.
>
> **Response:** These technical assumptions are in fact very mild. Part (1) simply suggests that the arrival rate $\lambda_1$ is upper bounded by some large constant when the utilization factor $\rho_1 =\lambda_1 /\mu_1$ is close to unity (in that most practical systems are in the so-called ``heavy-traffic'' regime where the arrival rate is close to the service rate). Part (2) is guaranteed by a sufficient condition $\lambda_1/\lambda_2 \ge \mu_1/\mu_2$, which means that the ratio of change in arrival rates (with respect to price) exceeds the ratio of change in service rates. This is indeed the case in most practical systems since price tends to affect demand rates much more than service rates.
>
> >**Comment 6.** There is some difficulty in understanding the process of demand-price the author(s) have in mind. Are prices computed for "resources" and then advertised for "services" which use these resources in a linear fashion? Also the following statements need clarification.
>
> **Response:** The prices are only advertised for products/services, i.e., $\mathbf{x} = (\{x_j: \ j \in \mathcal{J}\})$ and $\mathcal{J}$ is the set of products. Customers pay (a lump sum) $x_j$ to the firm for product $j$. We present this problem description in the first paragraph in Section 2 in the revised manuscript. We can think of selling products as selling a bundle of resources. We give each bundle a price (typically in practical application, the bundle price is less than the sum of individual resources). Again, we do not price resources nor do we add them in a linear fashion. This is the typical setting in the network revenue management literature (see the seminal paper by Gallego and Van Ryzin (1997)).
>
> >**Comment 7.** Line 68: In the first interval, the system completes serving the previous customers and reaches the steady-state under the new price. 69. Q: Only those customers? Any new customer arriving in the next interval are queued or are they rejected and cleared from the system until new prices are advertised?
>
> **Response:** We believe there are some misunderstandings here. We firstly answer your questions and then provide detailed explanation. (1) All the old customers and some newly arrived customers finish their services during the first interval in each batch with high probability. (2) Newly arrived customers are not rejected or manually held. They will use the service immediately if there are available resource and they will only be queued if there is no available resource.
>
> Imagine a single-server queue under a given price $p$. The customers arrive to this queue according to a Poisson process with rate $\lambda_p$, i.e., the time between two consecutive arrived customers is following an exponential distribution with mean $1/\lambda_p$. In this case, after some transient period (i.e., let the queueing system evolve), the queue reaches steady-state. Steady-state means that the system is in equilibrium, the distribution of the number of customers within the system remains unchanged over time (with arrivals, waiting, and departures continuously happening).
>
> The problem becomes much more challenging when we post a new price $p'$. As soon as we post a new price $p'$ at the beginning of batch, the arrival rate changes from $\lambda_p$ to $\lambda_{p'}$. Newly arrived customers are not rejected or manually held. They will use the service immediately if there are available resource and they will only be queued if there is no available resource.
> There could be old customers (with old service rates) in the system while new customers (with new arrival and service rates) can arrive at any time stamp in the future. A newly arrived customer can also finish using the product earlier than an old customer and thus the system can be very complicated. Thus, we propose to skip analyzing this complicated time (or transient time) but instead to bound the length of this transient time. Equivalently, we have to wait until the previously arrived customers under price $p$ to leave the system and the new system with new arrivals under price $p'$ reaches steady-state. That is why we have two intervals in each batch. The first interval of length $O(\tau)$ is used to ensure that the previous customers under price $p$ depart and the new system under price $p'$ reaches steady-state (or equilibrium). Not only the old customers have finished their services, but some new customers also finished their services when the system gradually reaches the steady-state. Then in the second interval, we can compute the equilibrium revenue under price $p'$, which is used to analyze regret. We never reject any customers and new customers are coming under $p'$ and potentially queued (if there are no available units) as soon as $p'$ is announced.

---

> > ### Comment · Reviewer_dRYY · 2022-08-03
> > **Part 2**
> >
> > Response to Comment 5.  "... which means that the ratio of change in arrival rates (with respect to price) exceeds the ratio of change in service rates."  So it seems the authors agree this is a technical condition which has not been sufficiently motivated by what it means in a real cloud setting.  It generally helps to explain why an assumption is even needed in the first place and if no natural reason exists, it could be stated that this is a technical assumption needed for proof of such and such a theorem.  Right now these assumptions are only stated in technical terms without an explanation why.  "This is indeed the case in most practical systems since price tends to affect demand rates much more than service rates."  And the evidence for this?
> >
> > Response to Comment 6:  OK, good, so it is only services that are priced for customers not resources.  This wasn't very clear to me.  And just for the record, the only example Gallego and Van Ryzin (1997) give for real life demand-price relationship is for air-fares which is remote from the cloud resource setting.  But let's just say OK to this.
> >
> > Response to Comment 7: I'm still not sure I'm following the syncopated stages considered in this model and why.  But let's say OK to this also.

---

> > > ### Author Response · Authors · 2022-08-08
> > > **Response to Reviewer dRYY's Reply (Part 2)**
> > >
> > > We value all your comments very much and we improve the presentation of this assumption following your suggestion. In the revised manuscript, we clearly state that the motivation of Assumption 5 is because it is used in the later proof. We also provide problem-wise explanation of this technical assumption and provide guidance for practitioners to validate whether this assumption is satisfied.
> > >
> > > ``This assumption is used to justify Lemmas 3 and 4 (used in the proof of Propositions 3 and 4). These conditions are mild and easy to validate in practice in the following sense. 1) The difference in arrival rates between two prices is bounded (by a large constant). This can be easily satisfied by interpolating intermediate prices in the candidate price set $\mathcal{X}$. 2) If both the service and arrival rate under a price is higher than or equal to another price, the utilization factor in the former case must be greater than the latter squared. A simple sufficient condition to guarantee $\rho_1 \geq  \rho_2^2$ is $\rho_1 \geq  \rho_2$, which reduces to $\lambda_{1}/\lambda_{2} \ge \mu_{1}/\mu_{2}$, and thus the practitioners only need to validate that the ratio of change in arrival rates exceeds the ratio of change in service rates.''
> > >
> > > We hope that we have adequately addressed all your major comments and we appreciate if you can kindly consider the possibility of raising our score.

---

> ### Author Response · Authors · 2022-08-01
> **Response to Reviewer dRYY - Part 3**
>
> >**Comment 8.** Line 69: In the second interval, the system maintains the steady-state (exploiting the new price). Q: By "maintain" the author(s) mean the customers for the next batch wait long enough for the system go past the transient phase and enter equilibrium? What happens during this time to newly arriving customers? What prices are advertised during this time?
>
> **Response:** Roughly speaking, we break the time horizon into batches of exponentially increasing length. So for a time horizon $T$, we have $O(\log T)$ number of batches indexed by $1,2,\ldots, M$. We only post a new price at the beginning of each batch and use that price for the entire batch (hence at most $O(\log T)$ number of prices are posted throughout $[1,T]$). For each batch $m=1,\ldots,M$, we further break it into two intervals. By the end of the first interval (with length $O(\tau)$, all the previous customers from the old price finished services and the new system under new arrivals under the newly posted price reaches steady-state; then during the second interval, the system keeps in equilibrium state (and this allows us to extract steady-state revenue under the newly posted price). Again, only a single price is advertised in each batch, and for the second interval of the batch, the system keeps in the equilibrium state (where the first interval allows time for the system to stabilize under this newly posted price). We also answer your questions specifically here:
> 1) The phase ``maintain the steady-state" here means the stationary distribution of the number of customers within the system has been reached. With arrivals, waiting, and departures (continuously happening), the distribution of the number of customers within the system at any time remains unchanged.
> 2) We are not holding customers nor manually pushing them waiting. Customers start to use a service once there are available resource units.
> 3) We advertise only one price in each batch. Thus, the two intervals in one batch is under the same advertised price.
> 4) When a new batch starts, consider the system changes price from $p$ to $p'$. There are previously arrived customers, who have already started to use our units, and they will keep using resource units under price $p$. New customers will arrive following Poisson process with new rate $\lambda_{p'}$.
>
> >**Comment 9.** Line 70: This batched framework not only allows for system stabilization (under new prices) but also allows for infrequent $O(\log T)$ price changes, which is more implementable in practical settings.
>
> Q: Why is price change not $O(T)$ if $T$ is selected to be the time during which prices don't change?
>
> **Response:** As we discussed in the previous question, we break the time horizon into batches of exponentially increasing length (think of power of 2's). So for a time horizon $T$, we have $O(\log T)$ number of batches indexed by $1,2,\ldots, M$. We only post a new price at the beginning of each batch and use that price for the entire batch (hence at most $O(\log T)$ number of prices are posted throughout $[1,T]$).
>
> >**Comment 10.** Line 133: Customers arrive following a price-dependent multivariate Poisson process with a total rate $\alpha(x)$, which is unknown to the decision maker a priori.
>
> Q: Has demand-price being multivariate Poisson been observed in any real setting? Otherwise is this is just for mathematical convenience? This reviewer has not seen demand-price to be a Poisson-process before unless the mean demand value of the Poisson process is some (elastic/inelastic) function of resource price. Btw, demand is going to be a function of service price and not resource price and the relationship between service price and resource price has not been discussed, as far as this reviewer can see.
>
> **Response:** Yes, multivariate Poisson arrivals with price-dependent arrival rates have been the predominant assumption in the revenue management literature (see the seminal paper by Gallego and Van Ryzin (1997)). As you mentioned, the arrival rates (or mean demand rates) are price dependent! In our setting, there exists a mapping $\mathbf{x} \mapsto \alpha(\mathbf{x}), P(\mathbf{x})$ that gives rise to product demand rates. The decision maker does not know this mapping and has to learn it over time while maximizing the total revenue on the fly. Also, what is special about this paper is that we assume a nonparametric mapping, which gives the most general results.
>
> >**Comment 11.** Line 141: We use a $J$-dimensional vector. Q: An $I$-dimensional vector?
>
> **Response:** Thank you for spotting this typo and we have fixed it.

---

> ### Author Response · Authors · 2022-08-01
> **Response to Reviewer dRYY - Part 4**
>
> >**Comment 12.** Line 144: $\pi (y; t) \in \mathcal{X}$ for period $t$ when there are $y$ customers in the system at the beginning of period $t$, to maximize the expected cumulative revenue $Y^\pi$.
>
> Q: Why is price only a function of state ($y \in R_+^I$) at the beginning of each period? Does demand for each period arrive instantaneously at the beginning when resource (should be service) prices are announced and no more demand arrives till the next time period? Also, prices are stated to be for resources but customers ask for services so is the price of service/product simply sum of resource used? This is of course true of costs but need not hold for prices. If so, this needs to be explicitly mentioned.
>
> **Response:**
> Thank you for your questions. The policy $\pi (\mathbf{y}; t) \in \mathcal{X}$ here means at the beginning of each period, the firm can observe the utilization across servers and the remaining time towards the end of the operation horizon, then it will post a price accordingly. Under full information (i.e., known mapping $\mathbf{x} \mapsto \alpha(\mathbf{x}), P(\mathbf{x})$), the state $(\mathbf{y},t)$ is sufficient statistics (i.e., complete summary of the stochastic system). For optimal policy under the full information scenario $\pi^*$, it knows how customers will react to the price and will choose the best price to maximize the future revenue that the firm can collect towards the end of the operation horizon. Under incomplete information (i.e., unknown mapping $\mathbf{x} \mapsto \alpha(\mathbf{x}), P(\mathbf{x})$), for our learning policy $\pi^{\textrm{RNRM-UCB}}$, it will utilize the historical sale data to choose the price with the highest UCB of the revenue rate. If your question is about where the historical data is in the learning policy, then our answer is: it is not included in the state $(y; t)$ but is included in the policy mapping $\pi^{\textrm{RNRM-UCB}}:  (\mathbf{y}, t, f_t) \mapsto \mathcal{X}$, where $f_t$ includes all historical actions and observations (arrivals, departures, revenues) up to time $t$.
>
> The customers arrive sequentially and stochastically, not only at the beginning of one period. Once a new price is posted, customers will come following a multivariate Poisson process. For example, consider one specific product type $j$ and the customers arrive under Poisson process with rate $a_j$, then the time interval between two consecutive customer arrivals is following an exponential distribution with mean $1/a_j$.
>
> The price is set for each product/service, not for resource. If customer requests to use product $j$, then he/she will pay a lump sum cost $x_j$ to the firm. This problem description is presented in the first paragraph of Section 2. We also include a concrete example of this network revenue management problem in the revised paper.
>
> Please, do not hesitate to ask any other questions you may have about our work, we will be happy to answer them. Thank you for your time carefully reviewing our work.

---

### Official Review · Reviewer_Upo2 · 2022-07-08

**Rating:** 6
**Confidence:** 3
**Soundness:** 3 good
**Presentation:** 3 good
**Contribution:** 3 good

**Summary:**

This paper designs a batched bandit algorithm for the network revenue management problem with reusable resources. Given $I$ resources (or servers) with fixed capacities, some requests for $J$ types of products arrive one by one, and each product needs to be processed by a sequence of product-dependent servers. The arrival rate of customers and the service rate of each server depend on the posted prices of
using each server, but the mapping between the rates and posted prices is unknown. The goal is to determine the posted prices to minimize the regret between the revenue of online algorithms and that of dynamic optimal policy.



**Questions:**

Please respond to two major issues mentioned in the weaknesses section.

**Ethics Review Area:**

["I don’t know"]

**Strengths And Weaknesses:**

**Strengths:**
1) The paper studies a practically relevant and challenging online learning problem for network revenue management with reusable resources.

2) The theoretical results are solid and significant (the regret results for both UCB and TS are near-optimal).

3) The analysis is nontrivial and presented clearly.

**Weaknesses:**
1) The key contribution of this paper is to consider the reusable resources in the network revenue management setting. However, there is a lack of characterization and discussions about the impact of learning the unknown service rate (related to the reusable resources). In fact, there is no procedure to learn the service rate in the proposed algorithm and the numerical results just use a fixed given service rate. Thus, please provide a clear explanation of why there is no need to learn the service rate for this problem. If it is due to the assumptions (e.g., memoryless customer flow and stability) that oversimplify the current problem and the authors should provide formal discussions
on this to clarify the contribution of this paper.

2) In real-world applications, the price of a product is usually proportional to the duration of using resources. Thus, the authors should provide a concrete application example that can fit the proposed model (each customer needs to sequentially go through product-specified resources; the customer can wait when the resource is unavailable without costs; the price of the product is independent of the duration of using resources).

**Minor comments:**
1) In the problem formulation, there seems to be a default assumption that each product only requests one unit of its requested resource. Could this assumption be relaxed?

2) The regret result (in Theorem 1) is linear in the number of product types J. Could the authors provide any remarks on the optimality of the regret in terms of J?

3) Typo: line 141, page 3, J-dimensional vector -> I-dimensional vector

---

> ### Author Response · Authors · 2022-08-01
> **Response to Reviewer Upo2 - Part 1**
>
> We sincerely thank the reviewer for the positive assessment and also offering constructive suggestions. We have uploaded a revised manuscript to address your comments with highlighted colors.
>
> >**Comment 1.** The key contribution of this paper is to consider the reusable resources in the network revenue management setting. However, there is a lack of characterization and discussions about the impact of learning the unknown service rate (related to the reusable resources). In fact, there is no procedure to learn the service rate in the proposed algorithm and the numerical results just use a fixed given service rate. Thus, please provide a clear explanation of why there is no need to learn the service rate for this problem. If it is due to the assumptions (e.g., memoryless customer flow and stability) that oversimplify the current problem and the authors should provide formal discussions on this to clarify the contribution of this paper.
>
> **Response:** Thank you for raising this good question. Service rates play an interesting role in this revenue problem. On one hand, the service rate does not directly affect the revenue collected from one specific customer. In our problem setting, each customer pays a lump sum amount to the firm, which is only determined by which product/service this customer requests and is not affected by the service duration (see our concrete example in the revised manuscript - a cloud solution may be sold as a bundle of resources and received a lump sum). On the other hand, the service rate affects the availability of reusable resources and thus would affect how much revenue the firm can gain in unit of time from a set of customers, i.e., it would affect how many customers can be successfully served in unit of time.
>
> In fact, the collected revenue of this network system is already very challenging to analyze even with known service rates, because one occupied unit can be released at any time stamp and the system dynamics can become very complicated. In our work, we compute the revenue without learning the service rates by the following two steps. Firstly, with Assumption 4 (stability), we can derive a steady-state revenue rate of this network service system under price $\mathbf{x}$, which can be expressed as $\vec{\alpha}_{\mathbf{x}}^T \mathbf{x}$. Secondly, with Propositions 3 and 4, we can use this steady-state revenue rate to compute the cumulative revenue for the second interval of each batch with high probability guarantee. Note that since the steady-state revenue rate derived does not depend on services rate, our learning algorithm can effectively work even without fixed nor known service rates.
>
> Comment on Assumption 4. We respectfully argue that this stability assumption is not over-simplifying the problem (see, e.g., Chen et al. 2020, https://arxiv.org/pdf/2009.02911). This stability assumption is commonly used in queueing system literature. This assumption also meets the practical requirement of network revenue management study. Any unstable service system with a risk of system explosion is undesired.
>
> >**Comment 2.** In real-world applications, the price of a product is usually proportional to the duration of using resources. Thus, the authors should provide a concrete application example that can fit the proposed model (each customer needs to sequentially go through product-specified resources; the customer can wait when the resource is unavailable without costs; the price of the product is independent of the duration of using resources).
>
> **Response:** Thank you for your suggestion. We have provided a real-world application in the revised manuscript. Since it contains pictures and detailed descriptions, we would very much appreciate if you can quickly look at our example in Appendix B in the revised manuscript.
>
> >**Comment 3.** In the problem formulation, there seems to be a default assumption that each product only requests one unit of its requested resource. Could this assumption be relaxed?
>
> **Response:** Thank you for your question. Given the current analysis, we cannot relax this unit-usage assumption because of reduction to the regular Jackson Network. But your suggestion raised a very interesting research question of having a potential reduction to a ``compound Poisson'' variant of Jackson Network (where consuming multiple units of the same resource can be viewed as compound Poisson arrivals). We will definitely revisit this problem in the near future.

---

> > ### Comment · Reviewer_Upo2 · 2022-08-05
> > **Response**
> >
> > I appreciate the response of the authors. Overall, I think this is a solid result. Just a quick suggestion, I appreciate adding the real example in Appendix B. It is still unclear how/why the price is independent of the resource's duration. I suggest some clear discussions on this. Thanks.

---

> > > ### Author Response · Authors · 2022-08-08
> > > **Response to Reviewer Upo2 with New Payment Model and Technical Results**
> > >
> > > We very much appreciate your prompt and favorable reply.
> > >
> > > We apologize for the delayed reply, since we spent the past three days working out the details of an alternative payment model that is dependent on the total service duration. We echo with you that this is important especially in the cloud computing setting. Now we have completely solved this new payment model and delivered new algorithm and technical results (please kindly see the newly added Appendix F). Fortunately, the main theorem/result remains unchanged.
> > >
> > > The revised rebuttal version now contains two payment models, namely, the lump sum model (in the main text) and the service duration dependent model (in Appendix F). We also provide more justification for the lump sum model in Appendix B. We hope that this completely resolves your concern, and we appreciate if you can consider the possibility of raising our score.

---

> ### Author Response · Authors · 2022-08-01
> **Response to Reviewer Upo2 - Part 2**
>
> >**Comment 4.** The regret result (in Theorem 1) is linear in the number of product types $J$. Could the authors provide any remarks on the optimality of the regret in terms of $J$?
>
> **Response:** Indeed, our dependence on $X$ and $T$ is optimal but we cannot claim the same for $J$, as you pointed out. That said, we would like to mention that a related reference is Miao and Wang 2021 (https://papers.ssrn.com/sol3/papers.cfm?abstract_id=3948140) who gave an upper bound of $\tilde{O}(J^{3.5}\sqrt{XT})$ for the non-reusable resource setting. All the existing works exhibit $\textrm{poly}(J,I)$ dependence for the regret upper bounds. A tighter lower bound (beyond the usual $\Omega(\sqrt{XT}$) remains open in the literature. We revised our statements in the revised manuscript.
>
> {\bf Comment 5.} Typo: line 141, page 3, $J$-dimensional vector $\rightarrow$ $I$-dimensional vector
>
> {\bf Response:} Thank you for pointing this typo out and we have fixed it.
>
> Please, do not hesitate to ask any other questions you may have about our work, we will be happy to answer them. Thank you for your time carefully reviewing our work.

---

### Official Review · Reviewer_ThAu · 2022-07-12

**Rating:** 4
**Confidence:** 3
**Soundness:** 3 good
**Presentation:** 2 fair
**Contribution:** 2 fair

**Summary:**

This paper looks at a setting where a firm offers products that require a sequence of reusable resources to be used in order for a customer to obtain the product. More specifically, the set of $J$ products is given by $\mathcal{J}$, the set of resources $I$ is given by $\mathcal{I}$, where each resource $i \in \mathcal{I}$ has a capacity (in terms of customers it can service at a given time) given by $c_i$. Customers are serviced for resources at exponentially distributd times, and there is assumed to be a topological ordering of resource dependencies for all products (so that customers can enter the overall system to be serviced in an ordered fashion irrespective of the product desired). Most importantly, the designer can assign prices $x \in \mathcal{X}$ to each product, paid for by customers as a lump sum before being serviced, and such that the prices define the Poisson processes that dictate demand for each product, as well as internal transitions between resource types for customers.

Given this context, the system designer is faced with the objective of maximizing the cumulative revenue of the system, wherein they can choose to dictate prices for goods at any given time period. However, the designer does not know the dependencies on demand and internal resource transitions beforehand, therefore a problem of balancing exploration of good prices and exploiting prices found to be good historically is faced. The main benchmark for performance used is regret, which is natural for this setting, and hence the algorithms provided by the authors are in the nature of those used for multi-armed bandits (UCB and Thompson sampling).

To prepare the setting for these algorithms, the authors first provide a proxy for regret given by an LP relaxation of the optimal policy in hindsight (relaxed regret), and furthermore, since both their variants on UCB and Thomspon sampling require changing price over different time batches, it is important to bound the mixing time of the underlying Markov chain governing product demand and resource usage. The authors do this for the setting of the paper, and can bound the regret accrued when the underlying chain mixes between price changes.

Finally, the authors provide corresponding regret guarantees, and provide empirical evidence in line with their theoretical results.

**Questions:**

Are there natural functional relationships between prices and demand that can give rise to better regret guarantees?

**Limitations:**

The largest limitation is the lack of distinction with existing paper (as per the weaknesses section)

**Strengths And Weaknesses:**

Strengths:
-The model at hand is well-justified as a natural extension of existing work
-The coupling argument is novel, as mentioned in the paper, especially in the context of mixing times for Jackson networks more generally

Weaknesses:
-The biggest weakness in my opinion is the fact that the authors have not spent enough time clearly distinguishing the results of this paper to that of "Online Learning and Pricing for Service Systems with Reusable Resources" (https://papers.ssrn.com/sol3/papers.cfm?abstract_id=3755902). The setting and techniques (and overall writing) of the paper are very similar, albeit the paper is for the single resource scenario. I do believe that there are interesting differences to be highlighted (such as in the coupling argument for example), but this needs to be much more clear in the paper.

---

> ### Author Response · Authors · 2022-08-01
> **Response to Reviewer ThAu**
>
> We sincerely thank the reviewer for reading our manuscript and asking us to further differentiate our contribution from prior work. We have uploaded a revised manuscript to address your comments with highlighted colors.
>
> >**Comment 1.** The biggest weakness in my opinion is the fact that the authors have not spent enough time clearly distinguishing the results of this paper to that of ``Online Learning and Pricing for Service Systems with Reusable Resources'' (\url{https://papers.ssrn.com/sol3/papers.cfm?abstract_id=3755902}). The setting and techniques (and overall writing) of the paper are very similar, albeit the paper is for the single resource scenario. I do believe that there are interesting differences to be highlighted (such as in the coupling argument for example), but this needs to be much more clear in the paper.
>
> **Response:** First, we thank the reviewer for appreciating the novelty of our coupling arguments in Jackson Network. In the originally submitted manuscript, we spent one paragraph (towards the end of Page 7) to highlight some of the challenges in extending the single-resource single-product result of Jia et al. 2020 to the network setting. In the revised manuscript, we further clarify the technical novelties in getting the finite-sample bound on mixing times of the Jackson Network:
>
> ``The coupling argument of Jia et al. (2020) in the single-resource single-product setting cannot be directly applied to our network setting. First, in the network of M/M/c queues, the arrival and waiting processes depend on previously visited nodes, which is significantly more complex to analyze upon price changes. To resolve this challenge, we have to carefully construct virtual and alternative queues to explicitly account for such dependency. On a high level, the difficulty lies in that the original queues have a time-varying rate from an initial rate to the stable rate upon any price change. One could construct virtual queues that could clear all previously arrived customers at a specific time $\tau$, but it is far too strong for the coupling analysis. Instead, we construct alternative queues that start from zero initial rate and ramp up directly to the stable rate at a time stamp before $\tau$ so that the customer distribution matches the original queues in period $\tau$. Second, even ignoring the subtleties of time-varying rates (upon price changes), if one applied the coupling argument of Jia et al. (2020) in a brute-force way to every node in the network, the coupling probability would result in a undesirable exponential scaling in the number of products. We decompose the network into topological layers so that the resulting coupling probability only scales linearly in the number of products.''
>
> We think our contribution to the understanding of Jackson Network is substantial, since this seems the first finite sample bound for any open queueing networks, which could open many doors to other network learning problems.
>
> >**Comment 2.** Are there natural functional relationships between prices and demand that can give rise to better regret guarantees?
>
> **Response:** Thank you for your nice suggestion. We consider a non-parametric function mapping from $\mathbf{x} \mapsto \alpha(\mathbf{x}), P(\mathbf{x})$ (i.e., total demand rate and routing probability matrix). Individual product demand can be expressed as a product-form of the routing probabilities of the underlying Jackson Network. Our result is optimal in the dimensions of $X$ and $T$.  Perhaps a linear functional dependency could improve the regret on the $J$ dimension and we will definitely look into this.
>
> Please, do not hesitate to ask any other questions you may have about our work, we will be happy to answer them. Thank you for your time carefully reviewing our work.

---

### Official Review · Reviewer_orv1 · 2022-07-13

**Rating:** 5
**Confidence:** 4
**Soundness:** 3 good
**Presentation:** 3 good
**Contribution:** 2 fair

**Summary:**

The authors study the network revenue management problem with reusable resources where the price-dependent arrival rates of all products are unknown and need to be learned. They propose UCB and TS-based batched algorithms that separate the time horizon into successive batches and select a price for each batch. The batch design helps bound the loss due to the nonstationarity in Jackson Network. Based on that, they provide instance-independent regret bounds for the proposed algorithms. Numerical experiments validate the performance of proposed algorithms.

**Questions:**

See Weaknesses.

**Limitations:**

See Weaknesses.

**Strengths And Weaknesses:**

Strengths
1) The online NRM problem where multiple products share multiple resources is new. The connection between this problem and the Jackson Network is also interesting.
2) To bound the loss due to nonstationarity, which is a unique challenge in this problem, the authors propose batched algorithms. The regret analysis of proposed algorithms relies on a finite-time high probability bound on mixing times of the Jackson Network system, which is one of the main technical contributions of this paper.

Weaknesses
1) One of my main concerns is Assumption 5, which is essential for the proof of Proposition 4 and the regret analysis. First, there is no explanation of “utilization factor” $\rho_1, \rho_2$. Second, it is unclear why the difference in arrival rates between two prices can be bounded by a function of only $\rho_1$. In which case this is not true?
2) There is no lower bound result in this paper, so it is hard to say whether the regret bound is tight, especially for $J$. Also, in lines 73-74, it said “we need to ensure that the regret scales sublinearly in the number of products”, however, the final regret is actually $O(J\sqrt{XT})$, which seems to be inconsistent.
3) The design of the batched UCB and TS algorithms is somewhat standard, so IMHO, the main technical contribution is how to bound the loss of nonstationarity.

Minor Comment
1) It is hard to see how the prices of UCB and TS change in Figure 1(a). Maybe a logarithmic x-axis could help.

---

> ### Author Response · Authors · 2022-08-01
> **Response to Reviewer orv1**
>
> We sincerely thank the reviewer for her/his careful review of our manuscript. We have uploaded a revised manuscript to address your comments with highlighted colors.
>
> >**Comment 1.** One of my main concerns is Assumption 5, which is essential for the proof of Proposition 4 and the regret analysis. First, there is no explanation of ``utilization factor'', $\rho_1,\rho_2$ . Second, it is unclear why the difference in arrival rates between two prices can be bounded by a function of only $\rho_1$. In which case this is not true?
>
> **Response:** The definitions of $\rho_1$ and $\rho_2$ are given by  $\rho_1=\lambda_1/\mu_1$ and $\rho_2=\lambda_2/\mu_2$ and we added them in the revised manuscript. Since $\lambda_1 \ge \lambda_2$ and $\lambda_2$ can be arbitrarily small, we can readily view this assumption as $\lambda_1 \le \frac{\rho_1}{-3e\log \rho_1}$. That is, the arrival rate $\lambda_1$ is upper bounded by some large constant when the utilization factor $\rho_1 =\lambda_1 /\mu_1$ is close to unity (in that most practical systems are in the so-called ``heavy-traffic'' regime where the arrival rate is close to the service rate).
>
> >**Comment 2.** About the lower bound dependence on $J$.
>
> **Response:** Indeed, our dependence on $X$ and $T$ is optimal but we cannot claim the same for $J$, as you pointed out. That said, we would like to mention that a related reference is Miao and Wang 2021 (https://papers.ssrn.com/sol3/papers.cfm?abstract_id=3948140) who gave an upper bound of $\tilde{O}(J^{3.5}\sqrt{XT})$ for the non-reusable resource setting. All the existing works exhibit $\textrm{poly}(J,I)$ dependence for the regret upper bounds. A tighter lower bound (beyond the usual $\Omega(\sqrt{XT}$) remains open in the literature. We revised our statements accordingly.
>
> >**Comment 3.** The design of the batched UCB and TS algorithms is somewhat standard, so IMHO, the main technical contribution is how to bound the loss of nonstationarity.
>
> **Response:** Indeed, our key contribution is to establish the first finite-sample bound on mixing times of Jackson Networks, which allows us to bound the loss of nonstationarity in a batched bandit framework. This seems the first finite sample bound for any open queueing networks, which could open many doors to other network learning problems. Another major contribution is the modeling of the network revenue management and its reduction to price-dependent Jackson Network, which is the first time such network structures are applied to a dynamic pricing problem with reusable resources.
>
> >**Comment 4.** It is hard to see how the prices of UCB and TS change in Figure 1(a). Maybe a logarithmic x-axis could help.
>
> **Response:** Thanks for pointing out this improving direction. We have updated the subfigure 1(a) in the revised manuscript. The price selections keep the same after Period 750 and thus we show the price selection only  for Periods 1 to 750.
>
> Please, do not hesitate to ask any other questions you may have about our work, we will be happy to answer them. Thank you for your time carefully reviewing our work.

---

> > ### Comment · Reviewer_orv1 · 2022-08-09
> > **Thanks for the response**
> >
> > Thank you for the response.
> >
> > I appreciate the response to Comment 1 which addresses my concern on Assumption 5.
> >
> > Overall, I think this paper has merits (especially the bound for the loss of nonstationarity), but the scope is a bit limited to learning in the Jackson Network. Thus, I'd like to keep my score.

---

### Meta-Review · Area_Chair_w81D · 2022-08-26

**Recommendation:** Accept
**Confidence:** Less certain

**Metareview:**

Executive summary:

The paper studies a price-based network revenue management problem with multiple products and multiple reusable resources, with an apriori unknown price-demand function. The authors approach this as a batched bandit learning problem, and their main result is an algorithm with cumulative regret \tilde{O}(J \sqrt(XT)) where J is the number of products, X is the number of prices, and T is the number of rounds. The dependence on X and T is best possible.

Discussion and recommendation:

This paper is a bit out of my comfort zone, so I am relying on the reviews for the decision. It seems that all reviewers appreciated the general model and the the theoretical results and experimental evaluation. Questions were raised regarding the various (seemingly) strong assumptions, but these were addressed in the rebuttal.

Weak accept.

**Award:**

No

---

### Decision · Program_Chairs · 2022-09-14

Accept